# On transformative adaptive activation functions in neural networks for gene expression inference

**Vladimír Kunc** ⬥ *, **Jiří Kléma**

Department of Computer Science, Czech Technical University in Prague, Faculty of Electrical Engineering, Prague, Czech Republic

* kuncvlad@fel.cvut.cz

## Abstract

Gene expression profiling was made more cost-effective by the NIH LINCS program that profiles only $\sim 1,000$ selected landmark genes and uses them to reconstruct the whole profile. The D–GEX method employs neural networks to infer the entire profile. However, the original D–GEX can be significantly improved. We propose a novel transformative adaptive activation function that improves the gene expression inference even further and which generalizes several existing adaptive activation functions. Our improved neural network achieves an average mean absolute error of 0.1340, which is a significant improvement over our reimplementation of the original D–GEX, which achieves an average mean absolute error of 0.1637. The proposed transformative adaptive function enables a significantly more accurate reconstruction of the full gene expression profiles with only a small increase in the complexity of the model and its training procedure compared to other methods.

## Introduction

Gene expression profiling is a great tool for medical diagnosis and deepening of the understanding of a disease (e.g., [1–4]). Despite a significant price drop in recent years, gene expression profiling is still too expensive for running large scale experiments. In order to economically facilitate such experiments, the LINCS program led to the development of the L1000 Luminex bead technology that measures the expression profile of $\sim 1,000$ selected *landmark genes* and then reconstructs the full gene profile of $\sim 10,000$ *target genes* [5]. The original profiling method used linear regression for the profile reconstruction due to its simplicity and scalability; it was then improved using a deep learning method for gene expression inference called D–GEX [6] which allows for a reconstruction of non-linear patterns.

The D–GEX is a family of several similar neural networks with varying complexity in terms of the number of parameters used for the gene expression inference. Our main contribution is that the D–GEX family performance can be further improved by using more suitable activation functions even while keeping the architecture unchanged.

In our paper, we first briefly present the use of artificial neural networks (ANNs) for gene expression inference and the usage of adaptive activation functions (AAFs) to improve the

**Data Availability Statement:** The generated data and the python implementation are publicly available at https://github.com/kunc/TAAF-D-GEX. The third-party data used for the training are publicly available at

https://cbcl.ics.uci.edu/public_data/D-GEX/. The authors had no special access privileges, and other researchers would be able to access this data in the same manner. The data used for the figures are within the manuscript and its Supporting information files.

**Funding:** This study was supported by the Czech Science Foundation (GACR - https://gacr.cz/en/) in the form of a grant awarded to JK (20-19162S) and the Research Center for Informatics (https://ec.europa.eu/) in the form of a grant awarded to JK (CZ.02.1.01/0.0/0.0/16_019/0000765). The funders had no role in study design, data collection and analysis, decision to publish, or preparation of the manuscript.

**Competing interests:** The authors have declared that no competing interests exist.

performance of ANNs. We then describe the data used and how they were processed before introducing the original D–GEX and, most importantly, our novel adaptive activation function. We also present several experiments to show that our proposed transformative adaptive activation function (TAAF) can significantly improve the performance of the original D–GEX.

## Artificial neural networks in biology

Artificial neural networks (ANN) represent a state-of-the-art approach in many fields (e. g. image classification, segmentation or reconstruction, natural language processing, and time-series forecasts), and biology is no exception (review e. g. [7–9]). The ANNs were used, for example, to analyze gene expression relationships [10], or for gene expression inference [6, 11, 12]. One such application is D–GEX [6], which infers a full gene profile using only $\sim 1,000$ selected *landmark genes*. The D–GEX family is made up of 9 different architectures. For technical reasons, D–GEX consists of two separate feedforward NNs having from one to three hidden layers—each having either 3,000, 6,000, or 9,000 neurons. Each network predicts only half of the target genes ($\sim 4,760$ genes) and is trained on a separate GPU. The NNs were trained using a standard back-propagation algorithm with mini-batch gradient descent with momentum and learning rate decay [6]. The initial weights were initialized using normalized initialization [13]. The error metric used was *mean absolute error* (MAE).

The original D–GEX was evaluated using data from three different sources—*GEO expression data* curated by the Broad Institute, *GTEx expression data* consisting of 2,921 gene expression profiles obtained using the Illumina RNA–Seq platform [14] and *1000 Genomes expression data* consisting of 462 gene expression profiles also obtained using the Illumina RNA-Seq platform [15]. The *GEO expression data* contained biological or technical replicates; the final dataset contained $\sim 110,000$ samples after removing these replicates. All three datasets were jointly quantile normalized and then standardized for each gene individually.

The D–GEX NNs were compared with linear regression and k–nearest neighbor (KNN) regression. The linear regression builds a linear model for each target gene, while the KNN regression finds $k$ closest expression profiles in the available data and returns the mean of the appropriate targets. The D–GEX NNs were found to perform superiorly on all three datasets. The $L_1$ and $L_2$ regularized linear regression performed similarly to non-regularized linear regression.

Another approach for gene expression inference appeared concurrently with our research—this approach uses Generative Adversarial Networks (GANs) for estimating the joint distribution of landmark and target genes [11, 12]. This approach resembles a two-player minimax game between two NNs—generative and discriminative models. Another approach based on the D–GEX, called L–GEPM, was presented in [16], where *long short-term memory* (LSTM) units were used. Yet another approach, albeit not based on neural networks, was presented in [17], where authors used the XGBoost algorithms for gene expression inference.

## Adaptive activation functions

The activation function introduces non-linearities to neural networks and is crucial for network performance [18]. Even though it might be suboptimal, the same activation function is usually used for the whole network or at least for all neurons in a single layer. Over the last few decades, there have been several attempts to use activation functions that might differ across neurons (e.g., [19–21]). The adaptive activation functions—i.e., functions that have a trainable parameter which changes their shape—have been receiving more attention recently (e.g., [21–24]) and might become a new standard in the field. The simplest forms just add a parameter to a particular neural network that controls one of its properties (e.g., slope) while the more

complex ones allow for the learning of a large number of activation functions (e.g., adaptive spline activation functions in [20]).

However, adaptive activation functions (AAF) might be very useful even in the simplest form with a single added parameter—an AAF called *parametric rectified linear unit* (PReLU) was used to obtain a state-of-the-art result on the ImageNet Classification in 2015, the first surpassing human-level performance [21]. The PReLU generalize the ReLU by adding a parameter that controls the slope of the activation function for negative inputs (the ReLU is constant at zero for negative inputs) that is learned with other weights: $f(y_i) = max(0, y_i) + a_i \, min(0, y_i)$, where $a_i$ is the optimized parameter. The *leaky rectified linear unit* (LReLU) [25] is essentially a PReLU but with the parameter $a_i$ fixed and not trainable. ELU (exponential linear unit) [26] is another popular activation function based on ReLU—it uses an exponential function for negative inputs instead of a linear function. An adaptive ELU extension *parametric ELU* (PELU) combined with mixing different activation functions which use adaptive linear combination or hierarchical gated combination of activation function was shown to perform well [27].

A sigmoid activation function with shape autotuning [28] is an example of an early adaptive activation function. The generalized hyperbolic tangent [29] introduces two trainable parameters that control the scale of the activation function. A more general approach was introduced in [30], which used networks with a trainable amplitude of activation functions; the same approach was later used for recurrent neural networks [31]. A similar approach to trainable amplitude and generalized hyperbolic tangent is the so-called *neuron-adaptive activation function* (NAF) [32]. Other examples include networks with an adaptive polynomial activation function [33], slope varying activation function [34] or back-propagation modification resulting in AAF [34, 35]. In parallel with our work, a neuron-wise and layer-wise adaptive activation function for physics-informed neural networks was presented in [36].

Another generalization of ReLU is the *adaptive piece-wise linear unit* (APLU), which uses the sum of hinge-shaped functions as the activation function [37]. An approach extending APLU is *smooth adaptive activation function* SAAF with piece-wise polynomial form and was specifically designed for regression and allows for bias–variance trade-off using a regularization term [23].

More complex approaches include spline interpolating activation functions (SAF) [20, 24, 38–40], which facilitate the training of a wide variety of activation functions. The *network in network* (NIN) [41], which uses a micro neural network as an adaptive activation function, represents a different approach. Yet another approach employs a gated linear combination of activation functions for each neuron; this allows each neuron to choose which activation function (from an existing pool) it may use to minimize the error [22]. A similar method uses just binary indicators instead of the gates [42]. The adaptive activation function might also be trained in a semi–supervised manner [43–45].

## Materials and methods

In order to examine whether our novel transformative adaptive activation function in D–GEX model could lead to lower error, we have used the very same data as in [6]. Furthermore, we have reimplemented the D–GEX and retrained it on the same data as the models with the novel TAAFs to ensure that the performance comparison indeed reflects only the influence of the usage of the novel transformative adaptive activation functions and nothing else.

### Data

We have used gene expression data from the Affymetrix microarray platform curated by the Broad Institute. It was provided by the authors of the original D–GEX [6] and contains

129,158 profiles, each consisting of 22,268 probes. The data are also available at https://cbcl.ics.uci.edu/public_data/D-GEX/. We have replicated the data pre–processing process presented in [6]—we have removed the biological and technical replicates and have used the same set of target and landmark genes. We have used 942 landmark genes to reconstruct the expression of 9,518 genes. This data was split into two datasets; the first dataset called *full dataset* and the second *heterogeneity–aware dataset*. The full dataset contains all data after preprocessing (126,102 samples) and it was split into a training, validation, and testing set (the training set has 88,256 samples while the validation set has 18,895 samples, and the testing set has 18,951 samples). The validation dataset was used for model selection and parameter optimization, while the testing set was used for reporting the performance of selected models based on out-of-sample data. The Heterogeneity–aware dataset contains a subset of the full dataset and was used for testing to determine whether the performance of the models on the full dataset might have been due to possible information leakage between training and testing splits.

**Heterogeneity–aware dataset.**   As in the original D–GEX paper [6], the data for most experiments were split into training and evaluation sets randomly; however, the data used contains different sets of samples that originated in the same experiment; thus such a split might have introduced bias to the reported results. To show that such bias, if present, is insignificant for our comparison, we have also run an experiment comparing our D–GEX reimplementation with D–GEX with TAAFs on a dataset, where the split was *GEO-* series aware (heterogeneity–aware dataset). We grouped the available samples from the full dataset by their *GEO-*series, if such a grouping was obtainable from the sample id. The we performed the split such that no group would have a sample in more than one split, which removed the potential information leakage between the splits. This resulted in a subset of the normalized data used consisting of 87,345 samples (the series information was not obtainable from the sample id for some samples) split into training (52,407 samples), testing (17,469 samples), and validation (17,469 samples) sets with no series overlaps. Since the lower amount of samples available for training might negatively influence the training and the resulting model performance and since it resembles the approach of [6], most of the experiments were done using the full dataset and the heterogeneity–aware dataset was used only to verify that the model performance is not due to the bias caused by information leakage between the sets.

## Data normalization

The data were preprocessed in the same manner as in [6] except for the last step—the scaling to a zero mean and unit standard deviation. Scaling each variable separately as in [6], however, removes the absolute differences in expression between individual genes. Moreover, it gives the same importance to all genes, including those whose expression is near noise levels, from the point of view of the error metrics. To keep the information about differences in expression levels, we scaled the data by transforming the whole data matrix to have zero mean and unit standard deviation without taking into account that there are different genes—thus, the more expressed genes will be proportionately more expressed even after the scaling. We believe that such scaling is more suitable in this case as the minimization of the error metrics during the fitting phase gives relatively higher importance to more expressed genes and less to the genes whose expression is near the noise level.

## D–GEX

D–GEX, as proposed in [6], is a feedforward neural network consisting of one to three fully connected hidden layers, each having the same number of neurons. The output layer consists of one neuron per target with a linear activation function. As in the original D–GEX, we have

split the set $\mathcal{G}$ of 9,518 genes into two random subsets, each containing half of the genes to enable learning on GPUs with smaller memory. A separate network was then trained using each of the sets, and the final reconstruction consists of outputs from both networks. The original D–GEX used *dropout* [46] as a regularization technique to improve the generalization of the network with three different dropout rates—0%, 10%, and 25%. Since the D–GEX with the 25% dropout rate had the best generalization [6], we have used only this rate for our experiments. All models were trained for 600 epochs (no improvement was observed near the end of the training). The performance of the model from each epoch was evaluated on the validation data, and only the best model from all epochs was used for further evaluation. The model optimization was done using the Nadam optimizer [47] with the learning rate $\mu = 0.0005$ and optimizer specific parameters $\beta_1 = 0.9$, $\beta_2 = 0.999$, and schedule decay $\eta = 0.004$; the batch size was set to 256 profiles.

## Model evaluation

To evaluate the model, we used the absolute error—first, we computed the *mean absolute error of prediction* $\text{MAE}_m(s)$ of model $m$ for sample $s \in \mathcal{S}$ over individual genes $g \in \mathcal{G}$ as in Eq 1 where $y(g, s)$ is the expression of gene $g$ for sample $s$ and $\widehat{y(g, s)}_m$ is the prediction of model $m$ for the same target.

$$\text{MAE}_m(s) = \frac{1}{|\mathcal{G}|} \sum_{g \in \mathcal{G}} |y(g, s) - \widehat{y(g, s)}_m|. \tag{1}$$

For further evaluation, we treat individual samples as independent (which is close enough to reality—our dataset probably contains small groups of samples that might be somewhat dependent, for example, having the same treatment, but it should be negligible for our size of dataset). Thus for pairwise comparison, we compare error metrics over individual samples and not over individual genes that have ties to each other. The overall performance of model $m$ is defined as:

$$\text{MMAE}_m = \frac{1}{|\mathcal{S}|} \sum_{s \in \mathcal{S}} \text{MAE}_m(s). \tag{2}$$

To estimate the distribution of the MMAE, we employ bootstrap over $\text{MAE}_m(s)$, i.e., we resample the set of samples with a replacement to get a new set $\mathcal{S}'$ which is then used for MMAE calculation in each bootstrap iteration. Pairs of models are not compared only in terms of MMAEs but also using pairwise differences. The mean difference of absolute errors $\text{MDAE}_{m_1,m_2}(s)$ for models $m_1$ and $m_2$ and sample $s$ is defined as:

$$\text{MDAE}_{m_1,m_2}(s) = \frac{1}{|\mathcal{G}|} \sum_{g \in \mathcal{G}} \left( |y(g, s) - \widehat{y(g, s)}_{m_1}| - |y(g, s) - \widehat{y(g, s)}_{m_2}| \right). \tag{3}$$

The $\text{MMDAE}_{m_1,m_2}$ is defined as:

$$\text{MMDAE}_{m_1,m_2} = \frac{1}{|\mathcal{S}|} \sum_{s \in \mathcal{S}} \text{MDAE}_{m_1,m_2}(s). \tag{4}$$

The pairwise nature of $\text{MMDAE}_{m_1,m_2}$ and its distribution allow for an accurate comparison of two models even though their MMAEs are very close, and their confidence intervals (CIs) estimated using bootstrap on MAEs are overlapping. The distribution is estimated using

bootstrap on $\mathrm{MDAE}_{m_1,m_2}(s)$ in a similar manner as distribution of $\mathrm{MMAE}_m$ is estimated using $\mathrm{MAE}_m(s)$.

To complement the model comparison based on MMDAEs, we have used the Student's paired t-test and the paired Wilcoxon rank test on MAEs of individual samples. These tests were used to test the hypothesis that the differences in MAEs for individual samples over all genes are significantly different.

## Proposed transformative adaptive activation function

We propose a novel adaptive activation function to further improve the original D–GEX. This proposal is based on an adaptive transformation of existing activation functions. The novel transformative adaptive activation function (TAAF) $g(f, y)$ introduces four new parameters $\alpha, \beta, \gamma, \delta \in \mathcal{R}$ per neuron, which transform the original activation function $f(y)$ (called *inner activation function* in the context of the TAAF):

$$g(f, y) = \alpha \cdot f(\beta \cdot y + \gamma) + \delta. \tag{5}$$

The output of a neuron with TAAF with inputs $x_i$ is:

$$\alpha \cdot f\left(\beta \cdot \sum_{i=1}^{n} w_i x_i + \gamma\right) + \delta, \tag{6}$$

where $x_i$ are individual inputs, $w_i$ are its weights, and $n$ is the number of incoming connections. If there is no unit $x_i$ (i. e. unit constant), then the parameter $\gamma$ is equivalent to the bias term of the neuron. The parameters are treated the same as other weights in the NNs and are learned using back-propagation and gradient descent—the only difference is that parameters $\alpha$ and $\beta$ are initialized to one and $\gamma$ and $\delta$ are initialized to zero in every neuron.

The motivation for the added parameters is that they allow arbitrary translation and scaling of the original activation function, and this transformation may be different for each neuron (i. e., each neuron has four additional parameters that define the TAAF for that neuron). Furthermore, such an adaptive activation function removes the need to have a linear activation function in the last layer for regression tasks as is usually done. The usage of the linear function in the last layer requires to have a full set of weights for the incoming connections just for the ability to scale the output to an arbitrary range while the proposed TAAF can do it with only 4 additional parameters.

The TAAF can also be viewed as a generalization of several existing adaptive activation functions—for example, the slope varying activation function [34] is the TAAF with adaptive parameter $\beta$, and frozen $\alpha = 1$, and $\gamma, \delta = 0$, or the trainable amplitude [30] is the TAAF with adaptive parameter $\alpha$, and frozen $\beta = 1$, and $\gamma, \delta = 0$. Other similar approaches also include parameters controlling slope but are focused only on a special, predefined function [29, 32] instead of allowing any activation function to be used as the inner function in the TAAF.

**TAAF as output layer.** It is standard practice to use an output layer with a linear activation function as the sigmoidal activation functions such as hyperbolic tangent and logistic sigmoid have limited ranges. The original D–GEX architecture is no exception and uses a linear output layer. This, however, is no longer necessary with the use of TAAF as the scaling and translation allow for an arbitrary range. The modified network architectures with TAAFs in the output layer (denoted TAAFo) enable better performance than those with a linear activation in the output layer by increasing the network capacity.

## Ensembles

Integrating multiple NNs (or other learners) into an ensemble very often leads to a better performance level than that of every single learner from the ensemble [48–51]. It is common practice to build ensembles of NNs even for quite complex NNs (e. g. [52, 53]). Ensemble usage is also a common practice when working with microarray data [54] (e. g. an ensemble of *support vector machines* was used in [55]). Further description of ensembles is out of the scope of this work; reviews are available in [50, 51, 56, 57].

We have evaluated ensembles consisting of different D–GEX architectures as the evaluation was without any significant computational overhead. Our ensemble selects a single D–GEX architecture as an expert for each gene based on one half of the validation data; then only this expert is used to predict the expression of the given gene—this leads to better performance if some NNs learned better prediction for some genes than the others. We have evaluated ensembles consisting of a maximum of four different architectures (a total of 984 ensembles for each activation function) based on models from Experiments 1—5 and selected those that performed best based on the second half of the validation data.

## Evaluation of the practical impact

Most of the evaluation of the models introduced focuses directly on the error of the gene expression inference; however, it is not entirely clear how lowering the error improves the accuracy of analyses applied to inferred data. To help clarify this, we show that the increased accuracy of the inference has both a statistically and practically significant impact on the accuracy of the differential gene expression (DGE) analysis.

There is no uniform phenotype annotation available for all of the samples. Therefore, we employed two distinct procedures to introduce a meaningful annotation for a limited sample subset at least. Firstly, we ran hierarchical clustering on 2,000 samples randomly sampled from the test data of the full dataset (i.e., unseen during the training of the model), then we selected two large and relatively distinct clusters each with more than 300 samples. In this way, we introduced two classes with naturally distinct expression profiles with a reasonable set of differentially expressed genes. Further in the text, we refer to these phenotypes as artificial. Secondly, we took the largest GEO- series (GSE2109) and made sure it was in the test data of the heterogeneity-aware dataset. We used the original classes from this series as phenotype information for another set of DGE analyses. Further in the text, we refer to these phenotypes as real.

Since obtaining microarray data for DGE is often very costly, we also show that the novel TAAFs improve performance even when using a low number of samples. We repeatedly sampled smaller datasets for different sample sizes where each half of the samples was from the same cluster and ran differential gene expression analysis using parametric empirical Bayes from the limma R package [58] on the ground truth data (the actual gene expression) and the gene expressions inferred by the evaluated models. The threshold $\alpha = 0.01$ for the adjusted p–value was used to determine the differentially expressed genes. For each model and each sampled dataset of a given size, we calculated the F1 score of the prediction of the differentially expressed (DE) genes compared to the DE genes from the ground truth data found for the same sample. Then we calculated the pairwise differences in the $F_{0.5}$, $F_1$, $F_2$ scores, accuracy, and Matthews correlation coefficient (MCC) for compared models. The differences of all scores ($F_{0.5}$, $F_1$, $F_2$ scores, accuracy, and MCC) were tested using the Wilcoxon signed-rank test.

## Implementation

The work was implemented in python 3, the neural networks were implemented using the NN library *Keras* [59] and the computational framework *Tensorflow* [60]. Other packages used

**Table 1. The MMDAE and its 95% CI estimated using bootstrap on samplewise MDAEs for the TAAF with tanh as inner activation function and classic tanh activation function for D–GEX with 25% dropout on the test data of the full dataset.**

| neurons | layers | MMDAE | 95% CI | |
|---|---|---|---|---|
| | | | TAAF tanh—tanh | |
| 3,000 | 1 | -0.016960 | -0.017064 | -0.016855 |
| | 2 | -0.008421 | -0.008472 | -0.008370 |
| | 3 | -0.015788 | -0.015867 | -0.015710 |
| 6,000 | 1 | -0.018504 | -0.018640 | -0.018366 |
| | 2 | -0.027463 | -0.027548 | -0.027376 |
| | 3 | -0.041951 | -0.042331 | -0.041683 |
| 9,000 | 1 | -0.020829 | -0.021007 | -0.020649 |
| | 2 | -0.049515 | -0.049631 | -0.049394 |
| | 3 | -0.063431 | -0.063633 | -0.063228 |

include *SciPy* [61], *scikit-learn* [62], *pandas* [63], and *NumPy* [64] for data manipulation and *Matplotlib* [65] and *seaborn* [66] for visualizations. The implementation is available at https://github.com/kunc/TAAF-D-GEX.

# Results

## Experiment 1: Usage of TAAFs

The goal of this and the following experiments is to establish the improvement as a result of using the novel TAAFs in models trained on the full dataset. First, we compare the original D–GEX architectures equipped with the hyperbolic tangent (tanh) activation function to architectures equipped with the novel TAAF with hyperbolic tangent as the inner activation function. The results are shown in Table 1, where the models are compared using the MMDAEs. The table shows the signed difference in absolute errors between the traditional hyperbolic tangent activation function and the adaptive activation function based on it—the novel transformative adaptive activation function was superior to the hyperbolic tangent activation function for all D–GEX architectures. Furthermore, the means (medians) of MMAEs for both models were significantly different using the paired Student's t-test (the Wilcoxon rank test) with $p < 0.0001$ for all D–GEX architectures tested.

## Experiment 2: Replacing tanh with sigmoid activation function

The performance of D-GEX with TAAF can be further improved by replacing the inner tanh activation function with a logistic sigmoid activation function. The comparison is shown in Table 2. Since tanh is just a simple transformation of the logistic sigmoid, this can be thought of as a different initialization of the TAAF parameters, namely $\alpha := \frac{1}{2}$, $\beta := \frac{1}{2}$, and $\delta := \frac{1}{2}$. The original D-GEX benefited from the sigmoid activation more (as shown in Table 3), which shows that it is much more sensitive to the activation function used and that using the TAAFs adds some robustness to the model over different parameterizations. Furthermore, even the version of D-GEX with sigmoid activation function significantly benefited from the use of TAAfs, as presented in Table 4.

## Experiment 3: TAAFs for capacity adjusted NNs

The TAAFs introduce four additional parameters per neuron, which increase the capacity of the neural network, and the improvement might possibly be caused by the increase in the capacity. Indeed, it seems that increased capacity helps D–GEX as the architectures with more

**Table 2. The MMDAE and its 95% CI estimated using bootstrap on samplewise MDAEs for the TAAF with tanh and sigmoid as inner activation functions for D–GEX with 25% dropout on the test data of the full dataset.**

| | | TAAF sigmoid—TAAF tanh | | |
|---|---|---|---|---|
| neurons | layers | MMDAE | 95% CI | |
| 3,000 | 1 | -0.005025 | -0.005120 | -0.004943 |
| | 2 | -0.017574 | -0.017725 | -0.017420 |
| | 3 | -0.010628 | -0.010732 | -0.010522 |
| 6,000 | 1 | -0.004390 | 0-.004550 | -0.004256 |
| | 2 | -0.009468 | -0.009560 | -0.009376 |
| | 3 | -0.002024 | -0.002214 | -0.001706 |
| 9,000 | 1 | -0.004043 | -0.004281 | -0.003853 |
| | 2 | -0.010392 | -0.010511 | -0.010271 |
| | 3 | -0.001712 | -0.001803 | -0.001615 |

**Table 3. The MMDAE and its 95% CI estimated using bootstrap on samplewise MDAEs for the sigmoid and tanh activation functions for D-GEX with 25% dropout on the test data of the full dataset.**

| | | sigmoid—tanh | | |
|---|---|---|---|---|
| neurons | layers | MMDAE | 95% CI | |
| 3,000 | 1 | -0.018551 | -0.018781 | -0.018345 |
| | 2 | -0.022399 | -0.022543 | -0.022253 |
| | 3 | -0.021952 | -0.022112 | -0.021786 |
| 6,000 | 1 | -0.018294 | -0.018676 | -0.017980 |
| | 2 | -0.033569 | -0.033709 | -0.033429 |
| | 3 | -0.038359 | -0.038547 | -0.038164 |
| 9,000 | 1 | -0.019727 | -0.020284 | -0.019274 |
| | 2 | -0.055361 | -0.055534 | -0.055192 |
| | 3 | -0.058344 | -0.058559 | -0.058129 |

neurons have a lower prediction error for the same number of layers. We have reduced the number of neurons in each layer in the D–GEX with TAAFs such that the total number of parameters is the same as in the original D–GEX with the same architecture. The number of removed neurons was always lower than 30 as the number of added weight per neuron is

**Table 4. The MMDAE and its 95% CI estimated using bootstrap on samplewise MDAEs for the TAAF with sigmoid as inner activation function and sigmoid activation function for D–GEX with 25% dropout on the test data.**

| | | TAAF sigmoid—sigmoid | | |
|---|---|---|---|---|
| neurons | layers | MMDAE | 95% CI | |
| 3,000 | 1 | -0.003434 | -0.003523 | -0.003330 |
| | 2 | -0.003595 | -0.003636 | -0.003555 |
| | 3 | -0.004464 | -0.004508 | -0.004419 |
| 6,000 | 1 | -0.004600 | -0.004748 | -0.004401 |
| | 2 | -0.003362 | -0.003407 | -0.003315 |
| | 3 | -0.005617 | -0.005674 | -0.005563 |
| 9,000 | 1 | -0.005145 | -0.005360 | -0.004860 |
| | 2 | -0.004546 | -0.004598 | -0.004491 |
| | 3 | -0.006799 | -0.006876 | -0.006724 |

**Table 5. The MMDAE and its 95% CI estimated using bootstrap on samplewise MDAEs for the TAAF with sigmoid as inner activation function and sigmoid activation function for D-GEX with 25% dropout on the test data of the full dataset.**

| | | TAAF sigmoid (reduced)—sigmoid | | | |
|---|---|---|---|---|---|
| neurons | layers | reduced | MMDAE | 95% CI | |
| 3,000 | 1 | 2990 | -0.003384 | -0.003476 | -0.003279 |
| | 2 | 2,997 | -0.003503 | -0.003543 | -0.003464 |
| | 3 | 2,997 | -0.004452 | -0.004493 | -0.004410 |
| 6,000 | 1 | 5980 | -0.004599 | -0.004746 | -0.004409 |
| | 2 | 5,997 | -0.003685 | -0.003732 | -0.003637 |
| | 3 | 5,997 | -0.005680 | -0.005734 | -0.005627 |
| 9000 | 1 | 8971 | -0.005130 | -0.005346 | -0.004849 |
| | 2 | 8,997 | -0.004139 | -0.004199 | -0.004077 |
| | 3 | 8,997 | -0.006811 | -0.006882 | -0.006740 |

The network with TAAF had a reduced number of neurons such that both networks had the same number of parameters—the final number of neurons in each layer is shown in the column *reduced*.

insignificant compared to the number of weights of incoming connections. The improvement of the reduced D-GEX with TAAFs was from 0.0034 to 0.0068 across different D-GEX architectures. The full comparison of the reduced D–GEXs with the adaptive activation function based on sigmoid and the original D-GEXs is shown in Table 5. We can observe that the reduction in the number of neurons had, as expected, only a small effect and that the network with TAAFs still significantly outperforms the original D-GEX.

## Experiment 4: Importance of individual parameters

The proposed TAAF introduces four additional parameters, and so far, the importance of individual parameters has not been established. Since the TAAF can be viewed as a generalization of several previously established AAFs, the performance increase compared to the sigmoid activation function might be due only to those parameters that were already established as beneficial (e.g., trainable amplitude [30]). Furthermore, since the proposed adaptive activation function is applied to the weighted sum of inputs in the neuron, parameter $\beta$ might seem redundant:

$$g(f, \vec{x}) = \alpha \cdot f(\beta \cdot \sum_{i=1}^{n} w_i x_i + \gamma) + \delta, \tag{7}$$

where $n$ is the number of inputs in the neuron, $x_i$ are individual inputs and $w_i$ are associated weights. This can be expressed without parameter $\beta$ if we define $u_i = \beta w_i$:

$$g(f, \vec{x}) = \alpha \cdot f(\sum_{i=1}^{n} u_i x_i + \gamma) + \delta. \tag{8}$$

While parameter $\beta$ seems to be redundant, redundancy by itself does not mean uselessness —in some cases, it can even improve the performance as shown in [67–69], where the authors introduced additional redundancy to neural networks to increase its performance, and in [70] where the authors discuss redundancy in a biological context with a connection to the artificial neural network architecture ResNet [71]. Another example of apparent redundancy can be found in overspecified neural networks—it was shown that overspecified wide networks

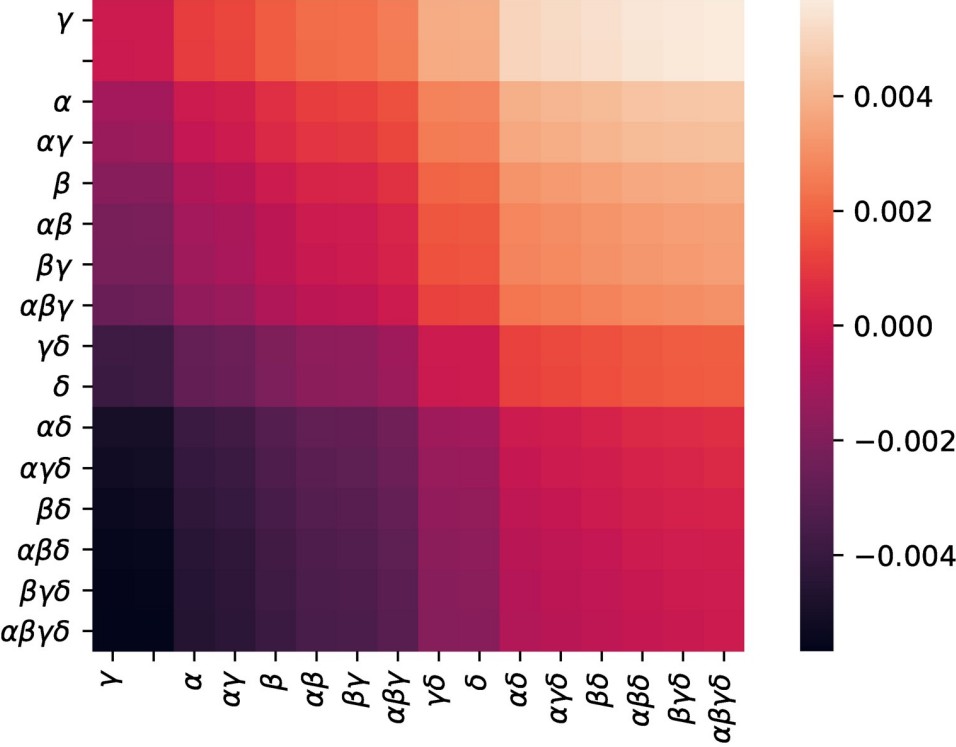

**Fig 1. MMDAE for different parameterizations.** The average mean difference in absolute errors for different model pairs. The model labels specify which adaptive parameters were used in the TAAF (e.g. $\alpha\beta$ means adaptive parameters $\alpha$ and $\delta$ were used).

simplify the optimization surface for optimizers in the sense that it is easier to reach a good optima [72–74].

Even though the redundancy represented by the $\beta$ parameter is different from some of the referenced examples, we empirically show that it improves performance and the improvement is statistically significant. The intuition behind redundancy in the form of additional parameters or overspecified networks is that "higher dimensions also means more potential directions of descent, so perhaps the gradient descent procedures used in practice are more unlikely to get stuck in poor local minima and plateaus" [72], which might be one of the reasons that the inclusion of the redundant parameter $\beta$ was empirically shown to be beneficial in our work.

To verify that all TAAF parameters improve performance, we trained neural networks with constrained TAAFs that had some of the parameters removed. We evaluated all 16 subsets of TAAF parameters (from the reduced TAAF equivalent to traditional sigmoid activation function to full TAAF with all four adaptive parameters) using three-layered D–GEX architecture with 6000 neurons in each layer. The networks with different subsets of TAAF parameters were pairwise evaluated based on MMDAE. Fig 1 shows the MMDAEs between all model pairs while Fig 2 shows whether model A (row) is significantly better than model B (column) based on the paired Wilcoxon rank test on samplewise MAEs at significance level $\alpha = 0.001$. The full TAAF is significantly better than all other combinations of parameters. This shows that the proposed TAAF with four parameters is the correct choice and that it outperforms the other adaptive activation functions it generalizes.

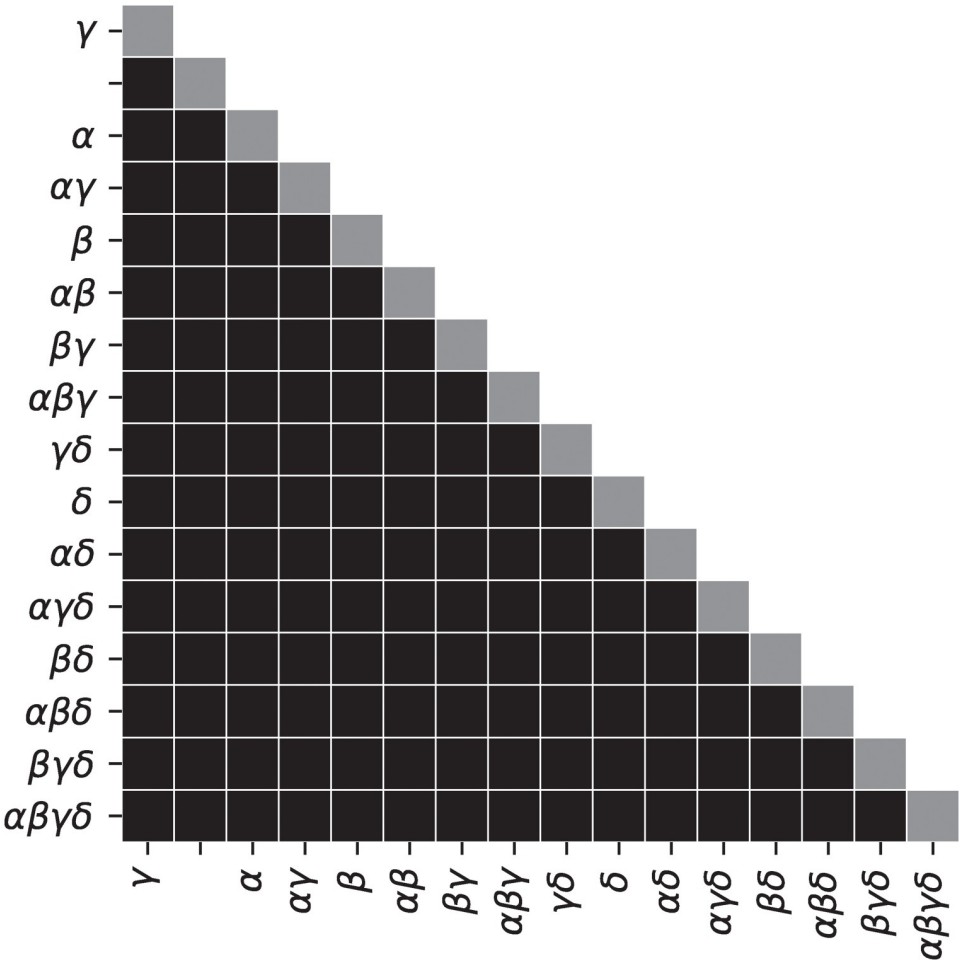

**Fig 2. Wilcoxon rank test for different parameterizations.** Testing for significant differences in samplewise errors—a cell is black if the model on the y-axis has a significantly lower MAE based on the paired Wilcoxon rank test at significance level 0.001 than a model on the x-axis. The model labels specify which adaptive parameters were used in the TAAF (e.g. $\alpha\beta$ means adaptive parameters $\alpha$ and $\delta$ were used).

## Experiment 5—TAAF in the output layer

The networks with TAAF do not require the output layer to contain a linear activation function for regression tasks as the TAAF allows for scaling and translation. Using TAAFs in the output layer might lead to better performance, as shown in Table 6, where networks with TAAFs in the output layer are compared with networks with a linear output layer. The usage of TAAFs in the output layer was beneficial for all architectures tested.

## Experiment 6—Heterogeneity-aware data sampling

We have also run an experiment comparing plain D-GEX and D-GEX with TAAFs (TAAFo) on the heterogeneity-aware data splits. The main focus of this experiment was to verify whether the possible bias due to information leakage between training and testing sets due to random splits in Experiments 1—5 is significant (if present at all).

Table 7 shows the relative comparison of plain D-GEX and D-GEX with TAAFs (TAAFo) for networks with sigmoid and hyperbolic tangent inner activation functions. The networks with TAAFs performed statistically significantly better than the plain D-GEX even on this

**Table 6. The MMDAE and its 95% CI estimated using bootstrap on samplewise MDAEs for the TAAF with sigmoid as inner activation function and 25% dropout on the test data.**

| neurons | layers | MMDAE | 95% CI | |
|---|---|---|---|---|
| | | TAAFo sigmoid—TAAF sigmoid | | |
| 3,000 | 1 | -0.000401 | -0.000472 | -0.000308 |
| | 2 | -0.001015 | -0.001091 | -0.000945 |
| | 3 | -0.001896 | -0.001951 | -0.001843 |
| 6,000 | 1 | -0.000679 | -0.000789 | -0.000531 |
| | 2 | -0.001654 | -0.001718 | -0.001591 |
| | 3 | -0.002474 | -0.002521 | -0.002428 |
| 9,000 | 1 | -0.000919 | -0.001075 | -0.000711 |
| | 2 | -0.001864 | -0.001935 | -0.001796 |
| | 3 | -0.001426 | -0.001477 | -0.001377 |

`TAAFo sigmoid` denotes a network that contains TAAFs in the output layer while `TAAF sigmoid` uses a linear activation in the output layer.

dataset, which had only ≈60% of the training samples compared to the whole dataset (some samples from the originally provided data were missing the GEO- id and could not be used). This result demonstrates that the above-mentioned bias does not affect the comparative analyses in our manuscript.

## Overall comparison

The best single network performs much better than our reimplementation of the original D–GEX—the MAE of the best network ($3 \times 9,000$ TAAFo with sigmoid) is 0.1340 (the 95% CI estimated over samples is [0.13316, 0.13486]) compared to D–GEX with tanh activation function with an MAE of 0.1637 (95% CI [0.16279, 0.16458]). Our proposed network performs better in 18,849 (99.75%) samples while worse in only 2 (0.001%) samples when the MAEs over genes for individual samples are compared using the paired Wilcoxon rank test at significance level $\alpha = 0.0001$.

All improvements to the original D–GEX are depicted in Fig 3, which shows the improvement of individual modifications. Fig 4 summarizes the individual improvements over the basic D–GEX with our proposed activation function. Table 8 shows the absolute performance of the top ten D–GEXs. We can observe that the TAAFs are much more robust with respect to their parameterization as there is only a small difference between using TAAFs with the logistic sigmoid and the hyperbolic tangent functions. In contrast, this difference is very large for

**Table 7. The MMDAE and its 95% CI estimated using bootstrap on samplewise MDAEs for the TAAF with sigmoid/tanh as inner activation function and sigmoid/tanh activation function for D–GEX with 25% dropout on the test set of the heteroginty-aware sampled data.**

| neurons | layers | TAAFo sigmoid—sigmoid | | | TAAFo tanh—tanh | | |
|---|---|---|---|---|---|---|---|
| | | MMDAE | 95% CI | | MMDAE | 95% CI | |
| 3,000 | 1 | -0.005339 | -0.005449 | -0.005221 | -0.005927 | -0.006007 | -0.005847 |
| | 2 | -0.006263 | -0.006329 | -0.006198 | -0.021192 | -0.021282 | -0.021103 |
| | 3 | -0.009114 | -0.009185 | -0.009042 | -0.019252 | -0.019341 | -0.019160 |
| 6,000 | 1 | -0.007214 | -0.007338 | -0.007082 | -0.005080 | -0.005174 | -0.004990 |
| | 2 | -0.005941 | -0.006012 | -0.005870 | -0.005400 | -0.005474 | -0.005326 |
| | 3 | -0.011624 | -0.011709 | -0.011539 | -0.010166 | -0.010256 | -0.010071 |
| 9,000 | 1 | -0.006664 | -0.006785 | -0.006539 | -0.005402 | -0.005515 | -0.005293 |
| | 2 | -0.006514 | -0.006589 | -0.006439 | -0.007455 | -0.007538 | -0.007368 |
| | 3 | -0.011349 | -0.011446 | -0.011250 | -0.011921 | -0.012038 | -0.011806 |

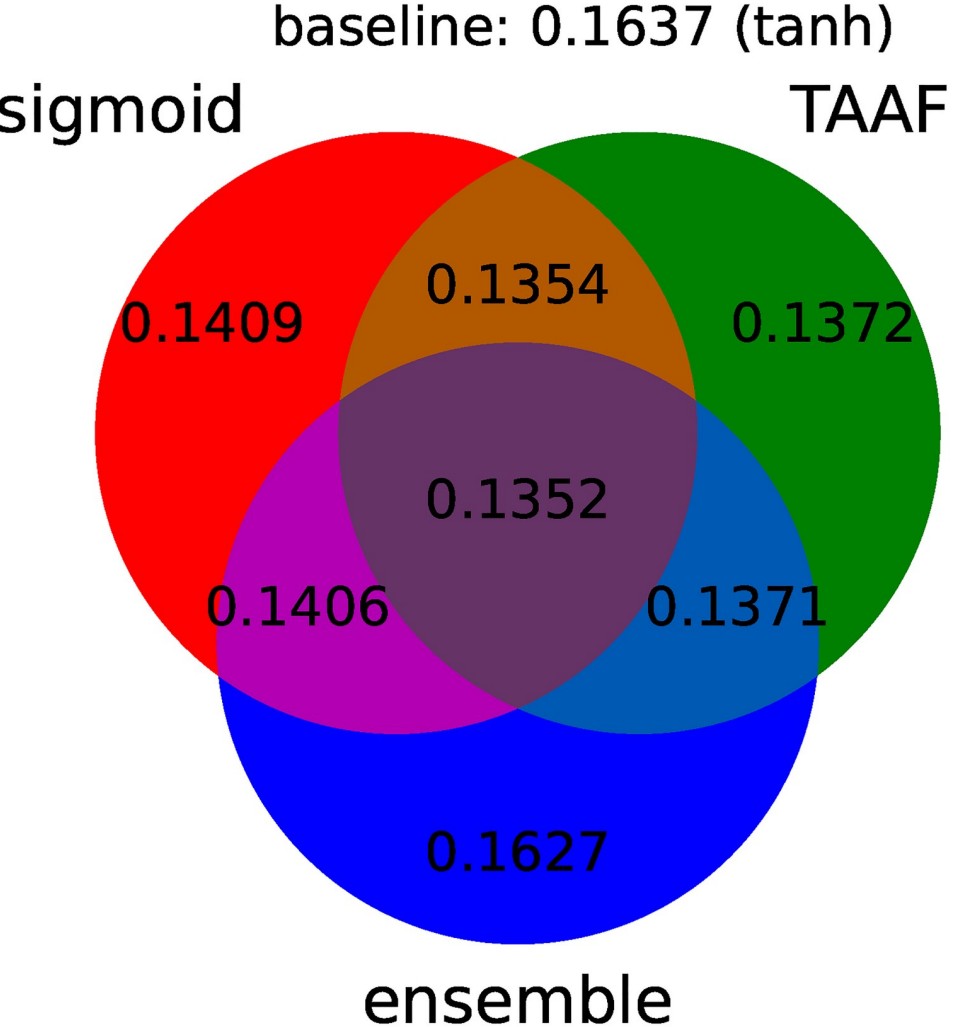

**Fig 3. Individual components of improvement.** A diagram depicting the performance for individual improvements over the standard D–GEX baseline with tanh activation function. The diagram shows the best MMAE over all D–GEX architectures for given approach trained on the full dataset.

the plain D-GEX, as shown in Fig 3. This robustness is important because the training of the networks is computationally costly, and thus, the parameter search possibilities are limited.

### Practical impact on differential gene expression analysis

To demonstrate that lowering the inference error has a practical impact on applied tasks, we ran differential gene expression analyses (DGEs) as described above. We started with the artificial phenotypes introduced by clustering, then we continued with the real phenotypes taken from the original annotation available for a particular data series.

   **Artificial phenotypes.**   First, we ran the DGE analysis using the best model from Experiment 1, which contained models trained on the full dataset, using the artificial phenotypes. The randomly sampled balanced datasets had sizes ranging 12—600, which is the usual sample size range for DGE analyses. The distribution of values and the pairwise differences of $F_1$ score is shown in Figs 5 and 6 for 5,000 repetitions for each sample size. The differences in all scores ($F_{0.5}$, $F_1$—$F_{10}$ scores, accuracy, and MCC) were statistically significant for all sample sizes tested when using the Wilcoxon signed-rank test as all the p–values were $<10^{-8}$.

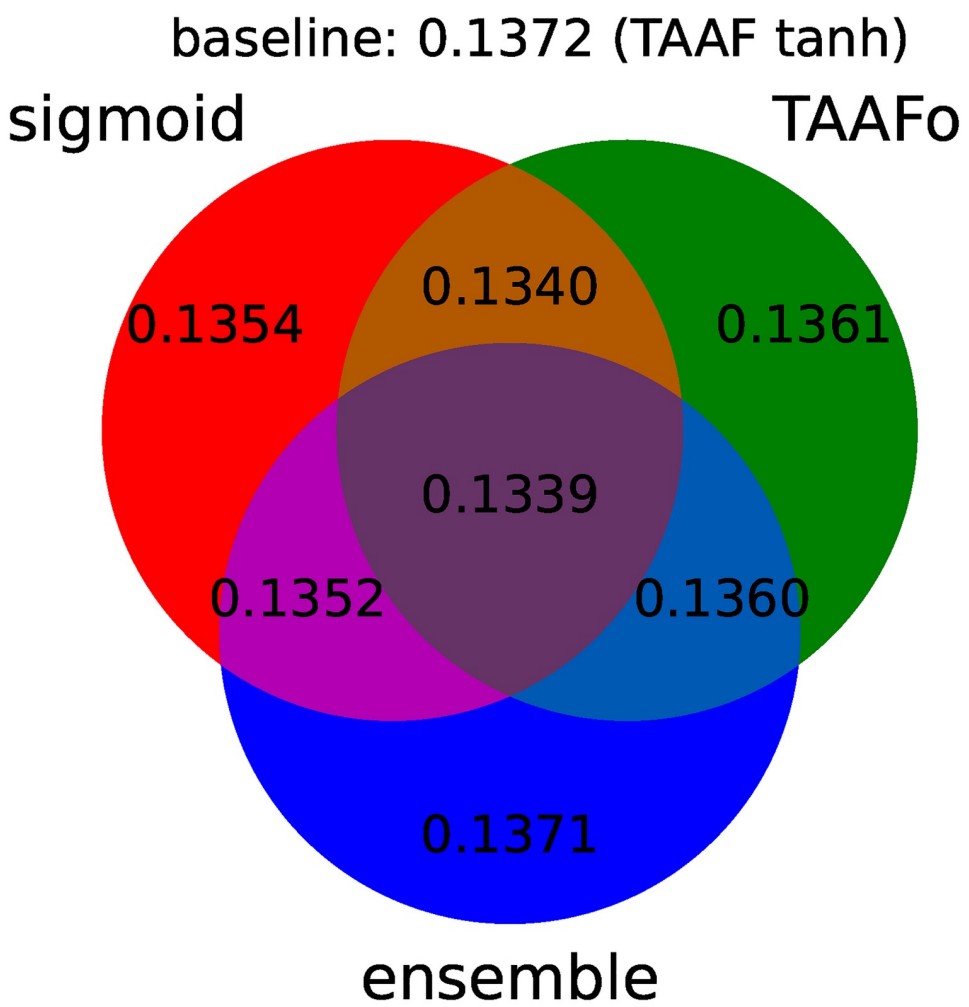

**Fig 4. Individual components of improvement including TAAFo.** A diagram depicting the performance for individual improvements over the D–GEX baseline already equipped with transformative adaptive activation functions. The diagram shows the best MMAE over all D–GEX architectures for given approach trained on the full dataset.

**Table 8. The 10 best D–GEX architectures in terms of MMAE on the test data of the full dataset.**

| rank | MMAE | neurons | layers | type | activation |
|---|---|---|---|---|---|
| 1 | 0.134015 | 9,000 | 3 | TAAFo | sigmoid |
| 2 | 0.134503 | 9,000 | 2 | TAAFo | sigmoid |
| 3 | 0.135430 | 8,997 | 3 | TAAF (reduced) | sigmoid |
| 4 | 0.135442 | 9,000 | 3 | TAAF | sigmoid |
| 5 | 0.136064 | 9,000 | 2 | TAAFo | tanh |
| 6 | 0.136367 | 9,000 | 2 | TAAF | sigmoid |
| 7 | 0.136774 | 8,997 | 2 | TAAF (reduced) | sigmoid |
| 8 | 0.136883 | 9,000 | 3 | TAAFo | tanh |
| 9 | 0.137154 | 9,000 | 3 | TAAF | sigmoid |
| 10 | 0.137189 | 6,000 | 3 | TAAFo | sigmoid |

25% dropout was used.

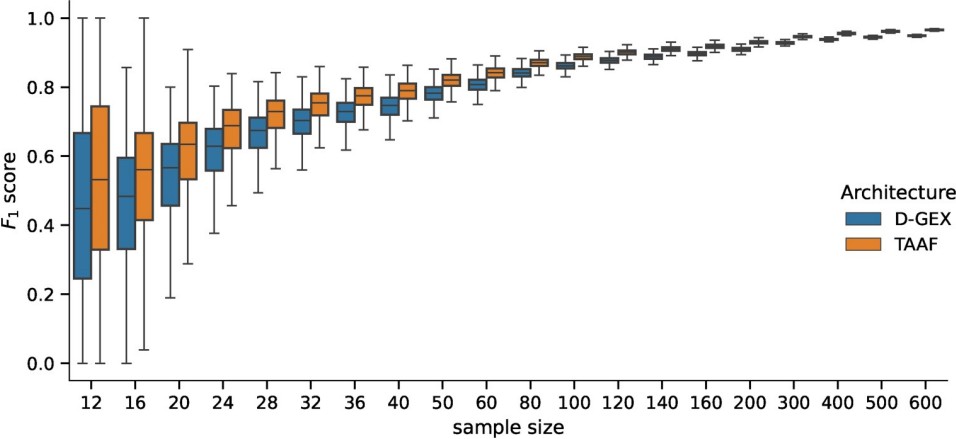

**Fig 5. Distribution of $F_1$ scores.** Distribution of the $F_1$ scores obtained by the D–GEX with TAAFs and the plain D-GEX of 5,000 repetitions for each sample size. The whiskers show the 10th and 90th percentiles.

We also analyzed how the candidates' ranking for differentially expressed genes differs between the ground truth data and the inferred data using both methods. For the second part of the analysis, we selected 100 candidates for differentially expressed genes (i.e., genes ranked 1—100 by their statistical significance) and compared their ranks when using the inferred data by both methods. The MAE of the rankings for both methods is shown in Fig 7, and the pairwise difference in Fig 8. The difference was statistically significant for all sample sizes tested when using the Wilcoxon signed-rank test with significance level $\alpha = 10^{-8}$. Both rankings were becoming more similar to the ground truth candidate ranking with increasing sample sizes in general (the selections of the first 100 candidates are becoming more and more conservative with increasing sample size). However, the candidate rankings produced using data inferred by D–GEX with TAAFs were closer to the ground truth rankings.

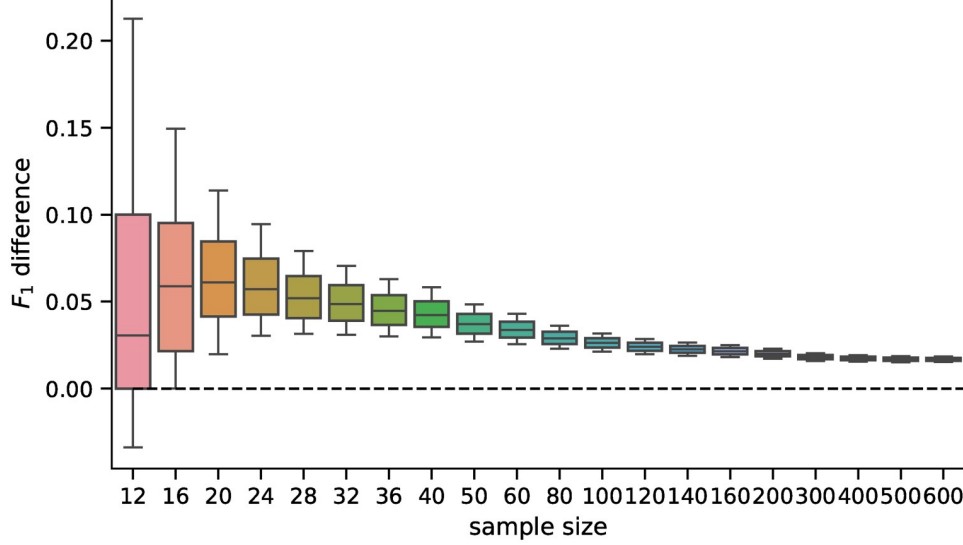

**Fig 6. Distribution of differences in $F_1$ scores.** Distribution of pairwise differences in the $F_1$ score obtained by the D–GEX with TAAFs and the plain D-GEX of 5,000 repetitions for each sample size. The whiskers show the 10th and 90th percentiles.

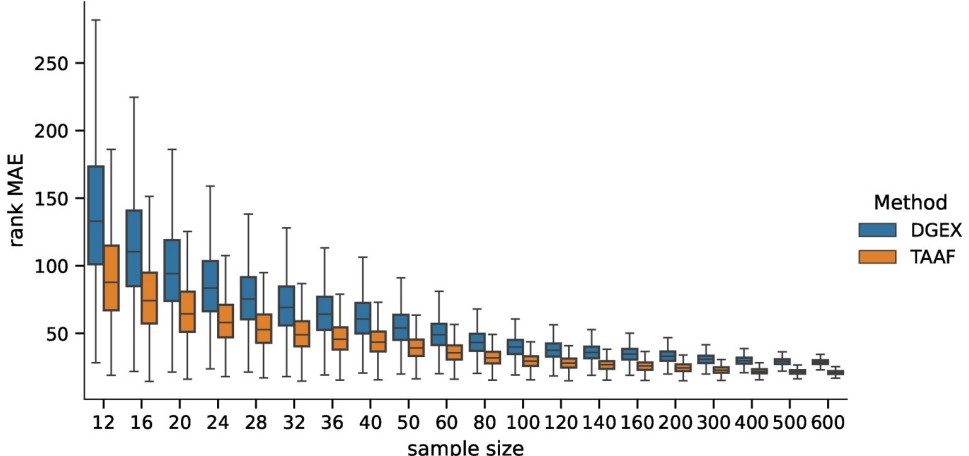

**Fig 7. Distribution of MAEs (first 100).** Distribution of the MAE on rankings obtained by the D–GEX with TAAFs and the plain D-GEX of 5,000 repetitions for each sample size for the first 100 candidates for DE genes selected on the sampled ground truth data. The whiskers show the 10$^{th}$ and 90$^{th}$ percentiles.

For the third part of the analysis, we selected candidates for differentially expressed genes as those genes for whose the p–value was lower than the significance level $\alpha = 0.05$. The MAE of the rankings for both methods is shown in Fig 9 and the pairwise difference in Fig 10. The MAE here is generally higher than in the case of the selection of only the first 100 candidates, but that is because the number of candidates selected with the threshold $\alpha = 0.05$ is higher and increases with the sample size. The increased accuracy of the D–GEX with TAAFs impacts the ranking for larger sizes as it obviously represents the expression data more faithfully, and thus the rankings are more similar to the ground truth data compared to the plain D–GEX.

**Real phenotypes.** We ran the DGE analyses as in the previous experiment again; however, this time using real phenotypes and model trained on the heterogeneity-aware dataset (the same as in Experiment 6) to show that the performance difference is not due to any potential

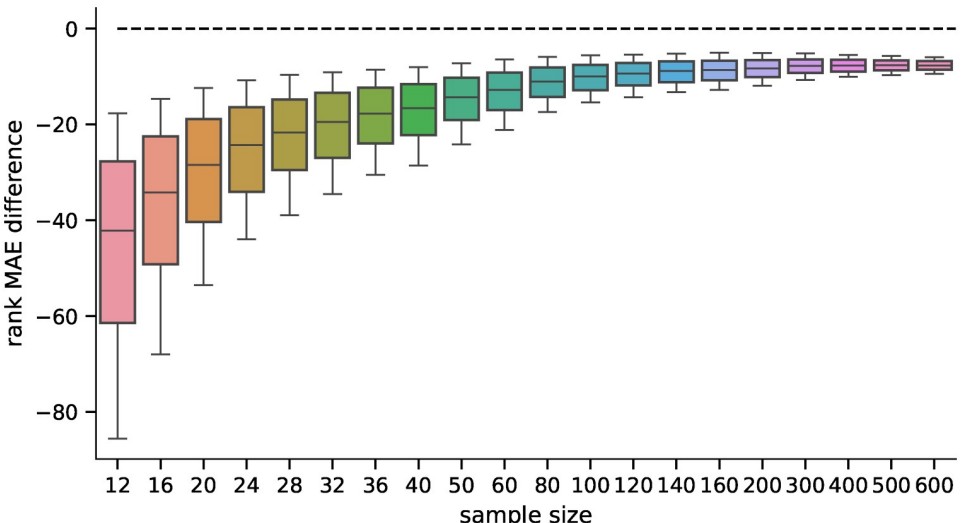

**Fig 8. Distribution of differences in MAEs (first 100).** Distribution of pairwise differences of the MAE on rankings obtained by the D–GEX with TAAFs and the plain D-GEX of 5,000 repetitions for each sample size for the first 100 candidates for DE genes selected on the sampled ground truth data. The whiskers show the 10$^{th}$ and 90$^{th}$ percentiles.

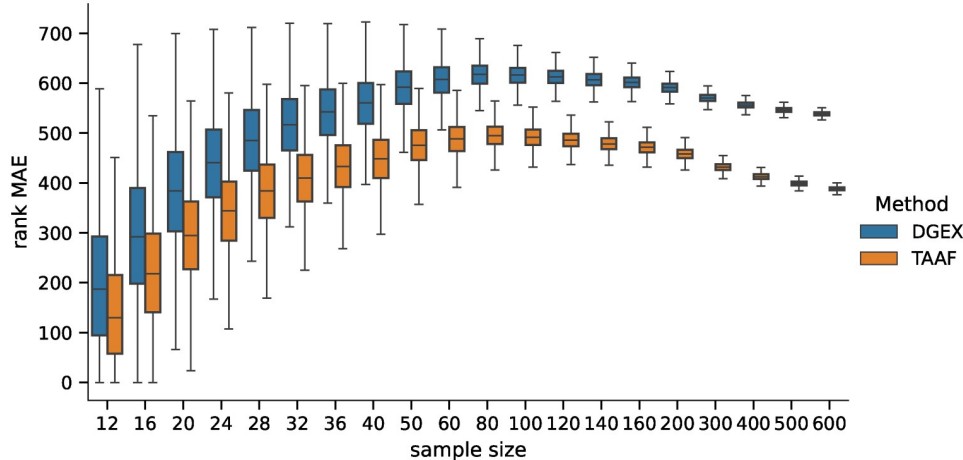

**Fig 9. Distribution of MAEs (p-value based).** Distribution of the MAE on rankings obtained by the D–GEX with TAAFs and the plain D-GEX of 5,000 repetitions for each sample size for the DE candidates selected on the sampled ground truth data at significance level $\alpha = 0.05$. The whiskers show the $10^{th}$ and $90^{th}$ percentiles.

information leakage to the test set and that the DGE performance difference is present even for actual phenotypes. We used $3 \times 9,000$ architectures with hyperbolic tangent and sigmoid inner activation functions for both the plain D–GEX and D–GEX with TAAFs. During the sampling procedure, we ensured that the GSE2109 series is present in its entirety in the test data and used it for the DGE analyses. The GSE2109 series consists of samples from different tissues; these tissues were used for phenotype classes for the DGE analyses. We selected classes that had more than 100 samples and ended up with six classes, as shown in Table 9. We then ran a DGE analysis for every pair combination, resulting in 15 analyses. The sampled balanced datasets sizes ranged from 12 to 200—400 depending on the class size for the tissue, the actual maximum sample size for a particular class is shown in Table 9. The sampled datasets were balanced, thus the smaller maximum sample size limit was used as the limit for the pair of classes.

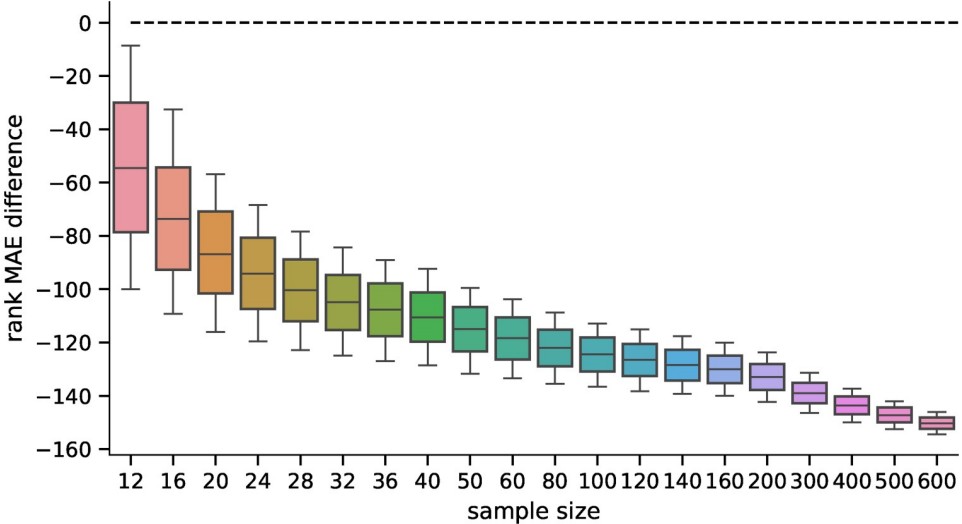

**Fig 10. Distribution of differences in MAEs (p-value based).** Distribution of pairwise differences of the MAE on rankings obtained by the D–GEX with TAAFs and the plain D-GEX of 5,000 repetitions for each sample size for the DE candidates selected on the sampled ground truth data at significance level $\alpha = 0.05$. The whiskers show the $10^{th}$ and $90^{th}$ percentiles.

**Table 9. The number of samples for each tissue in the GSE2109 series.**

| tissue | # samples | max sample size |
| --- | --- | --- |
| Breast | 351 | 200 |
| Colon | 292 | 200 |
| Kidney | 279 | 200 |
| Ovary | 198 | 150 |
| Uterus | 136 | 100 |
| Lung | 132 | 100 |

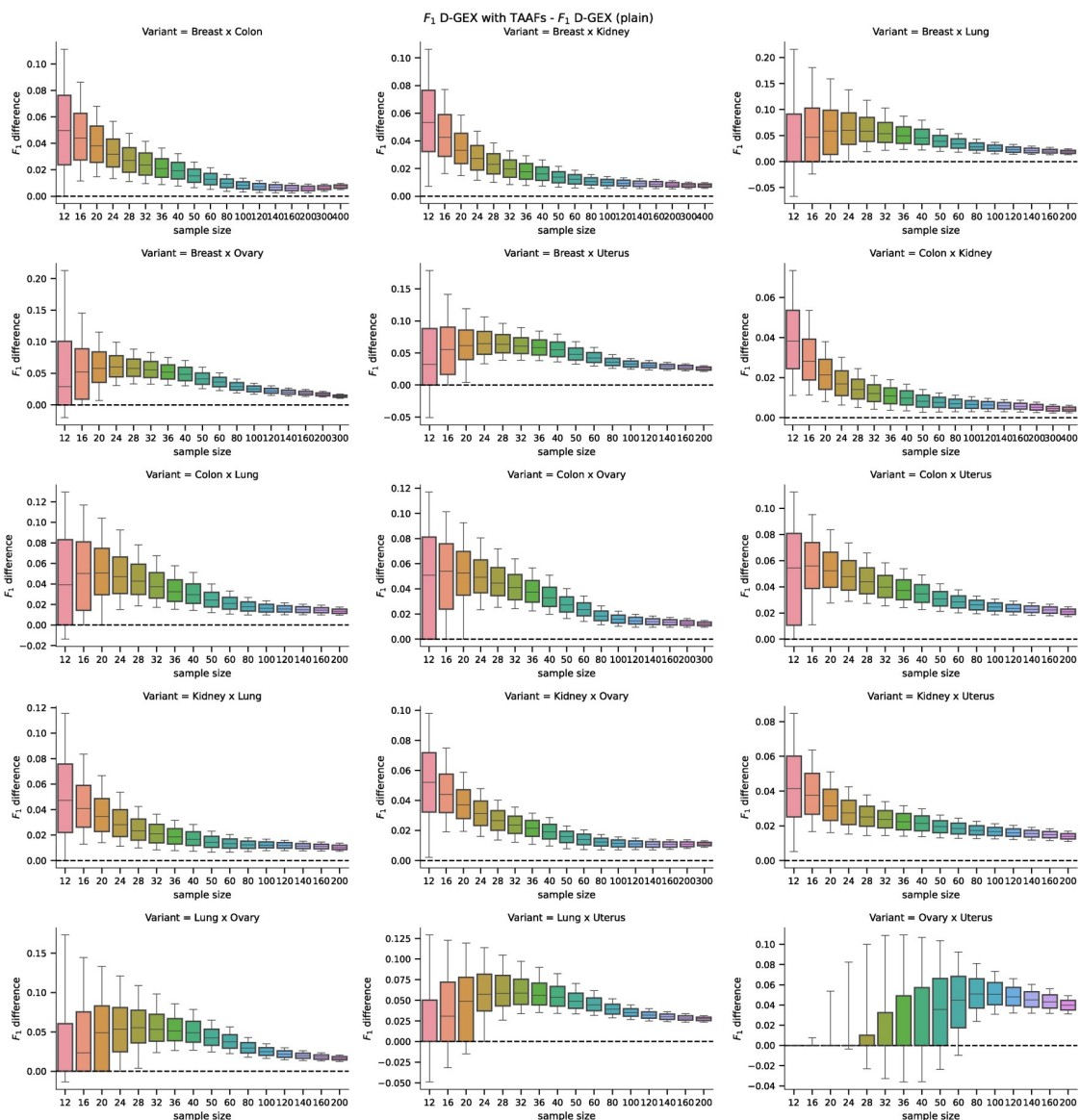

**Fig 11. Distributions of $F_1$ scores for the real phenotypes.** Distributions of the $F_1$ scores obtained by the D–GEX with TAAFs and the plain D-GEX of 5,000 repetitions for each sample size for different tissues. The whiskers show the $10^{th}$ and $90^{th}$ percentiles.

The distributions of values and the pairwise differences of the $F_1$ score are shown in Fig 11. The differences in the accuracy, $F_1$, $F_{0.5}$, $F_2$, and MCC scores were also tested using the Wilcoxon signed-rank test with significance level $\alpha = 10^{-8}$. The D-GEX with TAAFs statistically significantly outperformed the plain D–GEX for most of the tasks and sample sizes, detailed results are shown in Fig 12—the only exception is the *Ovary × Uterus* task, where both models performed very similarly for small sample sizes and no statistically significant performance difference was observed at the given significance level.

The analysis of the impact on candidate rankings for the differentially expressed genes was also run for all 15 tissue pairs. First, we analyzed the difference in the rankings of the first 100 candidate genes selected by the DGE using ground truth data. The results for individual tasks and sample sizes are shown in Fig 13. Second, we analyzed the difference in the rankings of the candidate genes whose p-value from the DGE on the ground truth data (for the particular sample) was above $\alpha = 0.05$ (leading to non-constant sizes of the candidate sets); the results are shown in Fig 14. The differences in MAE of the rank differences were tested using the Wilcoxon signed-rank test with significance level $\alpha = 10^{-8}$. The D-GEX with TAAFs was statistically significantly better for both candidate selection methods, all tasks and all sample sizes with p-value $<10^{-8}$. Therefore, it is safe to conclude that if there is some bias in the

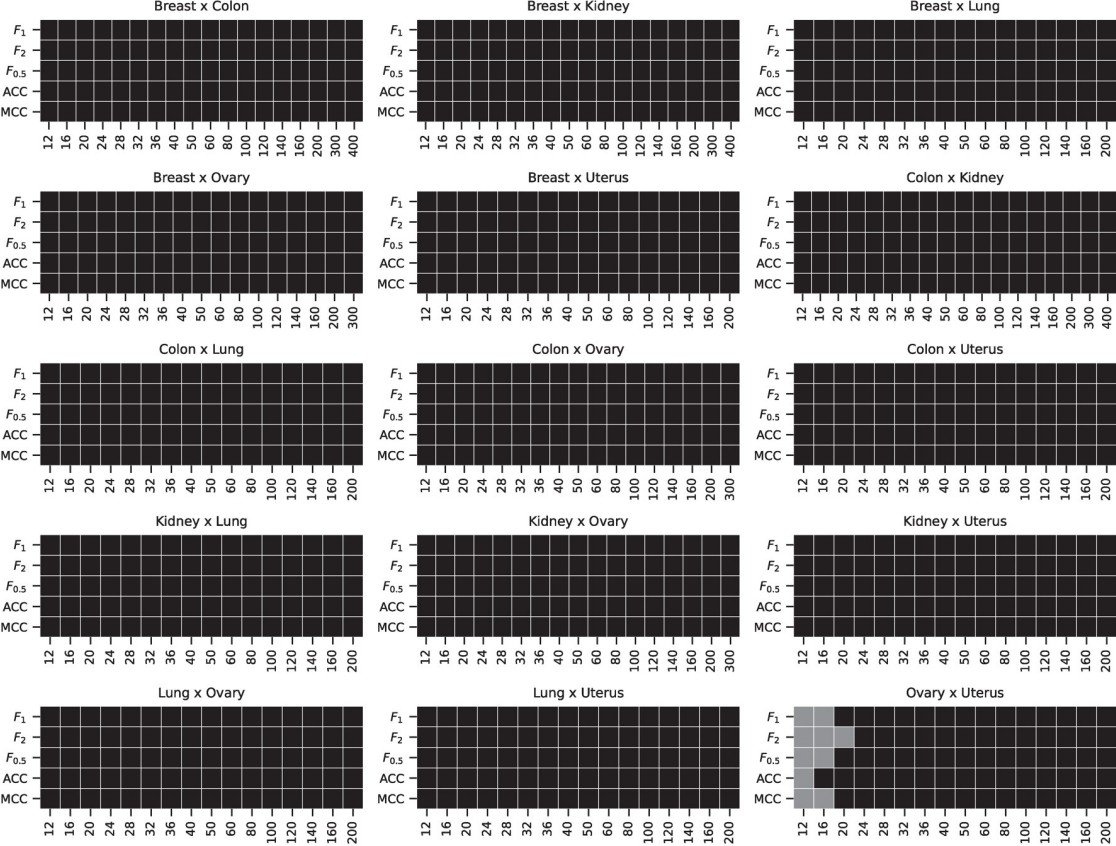

**Fig 12. Results of the Wilcoxon test for individual tasks pairs.** The results of the metrics pairwise comparison using the Wilcoxon signed-rank test with significance level $\alpha = 10^{-8}$. The cell is colored black if the D-GEX with TAAFs performance for the metric was statistically significantly better than the plain D-GEX, white if the plain D-GEX performed better and grey if no statistically significant difference was found.

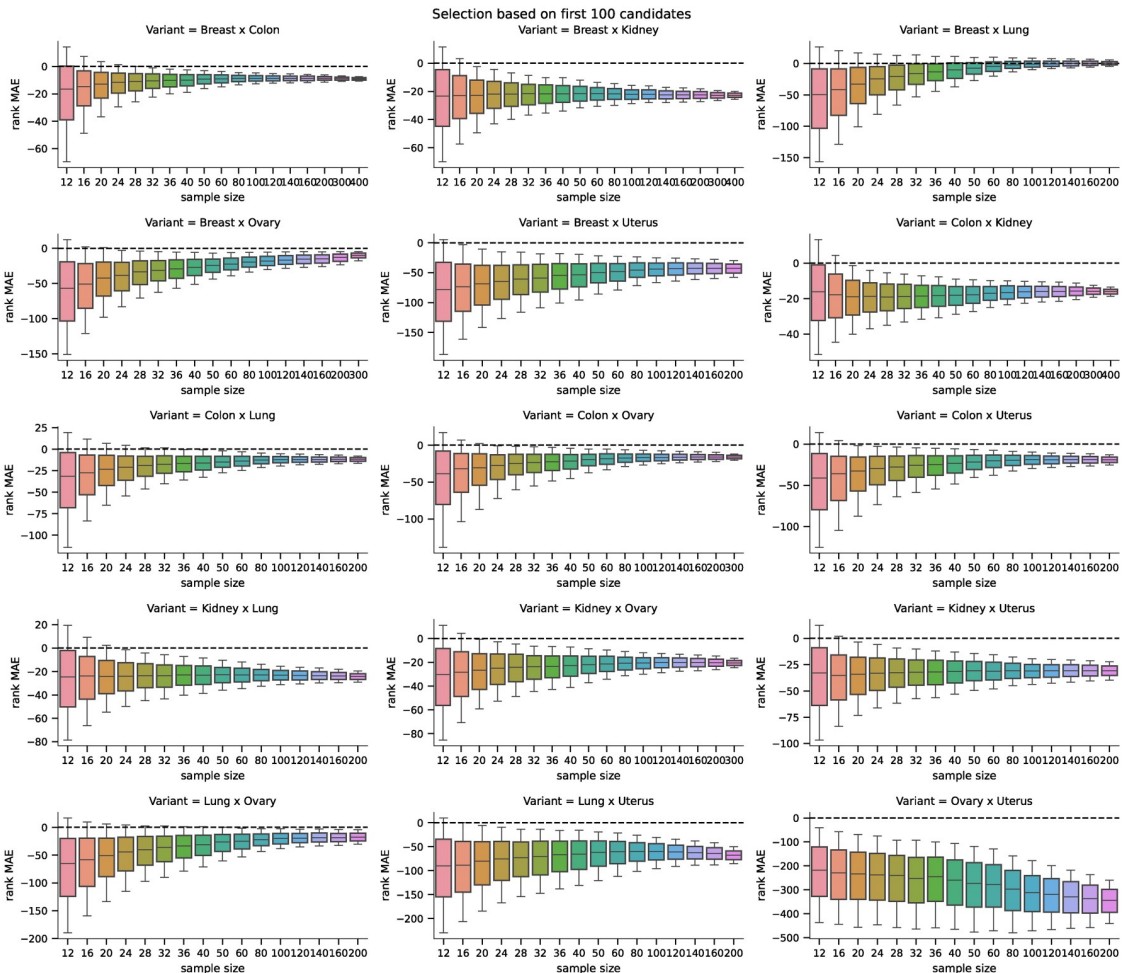

**Fig 13. Distributions of differences in MAEs (first 100) for the real phenotype.** Distributions of pairwise differences of the MAE on rankings obtained by the D–GEX with TAAFs and the plain D-GEX of 5,000 repetitions for each sample size for the first 100 candidates for DE genes selected on the sampled ground truth data for different tissues. The whiskers show the 10[th] and 90[th] percentiles.

performance of the models on the full dataset, it is not significant for the model comparison and the experiments and model trained on the full dataset are valid.

## Discussion

The experimental results show that the TAAFs lead to significant improvement in the gene expression inference task. We believe that this leads to new state-of-the-art results; however, this is rather unclear since we are using a different normalization technique than other works [6, 11, 12]. Our reimplementation of D–GEX has an MMAE 0.1637 while the best performing D-GEX with TAAFo has an MMAE of 0.1340; this means $1 - \frac{0.1340}{0.1637} \approx 18\%$ improvement and $1 - \frac{0.1361}{0.1637} \approx 17\%$ when using tanh for the inner activation function while the presented improvements using complex GAN approach have $1 - \frac{0.2997}{0.3204} \approx 6.5\%$ improvement [11] and $1 - \frac{0.2897}{0.3204} \approx 9.6\%$ in [12]. However, both TAAFs and GANs are not mutually exclusive and can potentially be used together to achieve even better performance. Furthermore, the TAAF

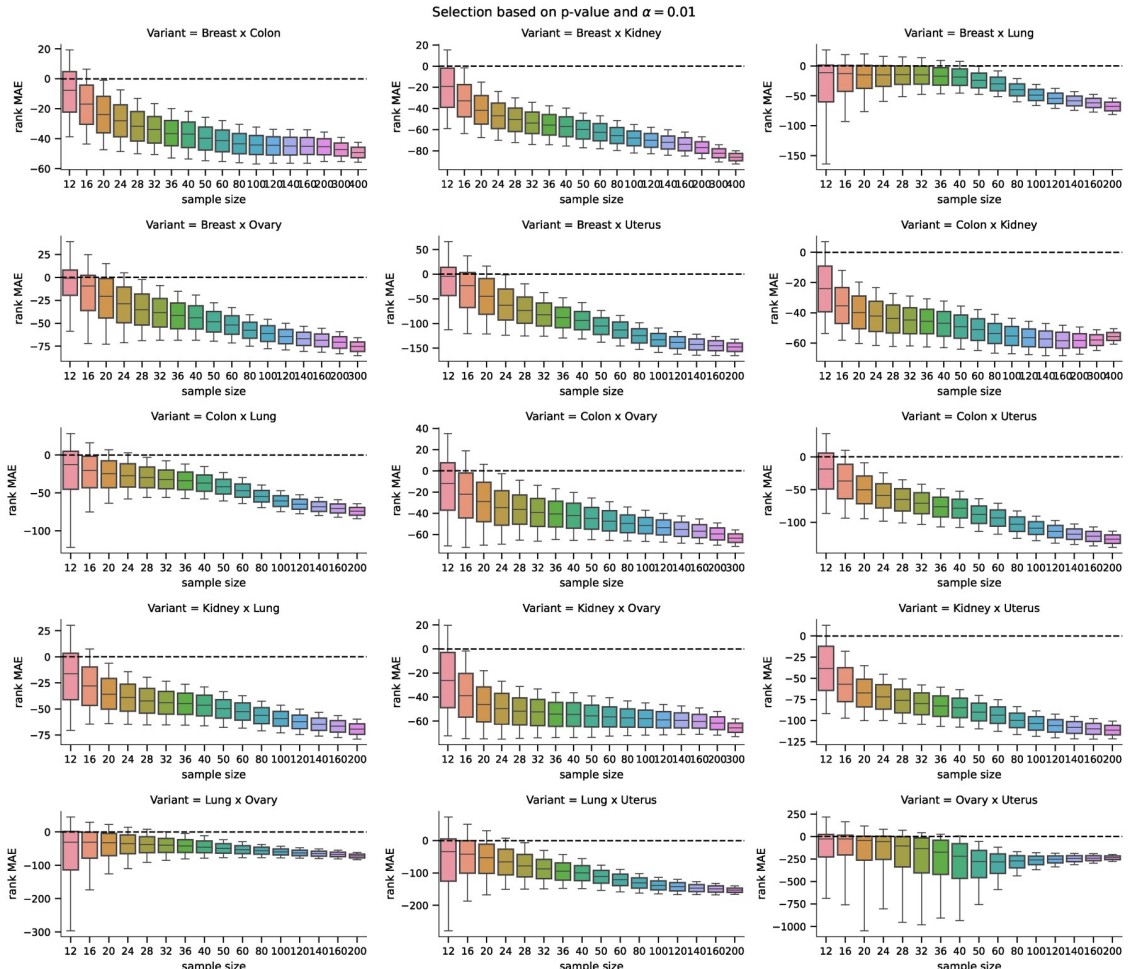

**Fig 14. Distributions of differences in MAEs (p-value based) for the real phenotype.** Distributions of pairwise differences of the MAE on rankings obtained by the D–GEX with TAAFs and the plain D-GEX of 5,000 repetitions for each sample size for the DE candidates selected on the sampled ground truth data at significance level $\alpha = 0.05$ for different tissues. The whiskers show the 10th and 90th percentiles.

approach is conceptually much simpler than the usage of GANs while reaching, at the very least, comparable performance.

Almost the same argument can be made with the L–GEPM tool from [16], which reached the error 0.3213 while their reimplementation of D-GEX reached an error 0.3448, which is $1 - \frac{0.3213}{0.3448} \approx 6.8\%$ improvement. The slight disadvantage of L-GEPM is using the LSTM units, which are much more complicated than neurons with the TAAFs. As with the GANs, the L–GEPM approach is not mutually exclusive from the TAAFs, and both can possibly be used together to achieve better performance.

Since the used data contain samples from many tissues and experimental conditions, it is possible that the actual quality of the inference differs based on the conditions and tissue as the model estimating procedure reflects the data used in the estimation. This limitation is unfortunately inherent to the data available and is present for all models present in the literature for the L1000 gene expression inference as all of them use the GEO data for model parameter estimation [6, 11, 12, 16]. While no quality guarantees for individual tissues and experimental

conditions are available, we have shown that our novel model performs generally better than the original D–GEX on available data.

## Conclusion

We proposed a novel transformative adaptive activation function (TAAF), which allows for scaling and translation of any other activation function. We evaluated the novel TAAF in the D–GEX settings and verified that it outperforms its non-adaptive counterparts. The proposed TAAF generalizes several existing adaptive activation functions [30, 34] and performs superiorly. Our three-layered D-GEX with TAAFo significantly outperforms the best D–GEX architecture with $p < 0.0001$ and improves the MAE by $\approx$18%.

Furthermore, we concluded that the sigmoid activation functions are the better choice for this task despite the hyperbolic tangent being typically more suitable and recommended in [18] as the D–GEX worked better with sigmoid both with and without the TAAFs compared to the tanh activation function. However, the difference was much smaller for the D–GEX with TAAFs—the TAAFs add robustness to the model.

We have also shown that lowering the error using TAAF has a practical impact on differential gene expression analysis. The improvement has shown statistically significant differences in the accuracy, Matthew's correlation coefficient and the F scores for both the artificial and real phenotypes. Eventually, the results reached proved to be unaffected by the potential learning biases resulting from the heterogeneity of the input dataset.

While we have used D–GEX model for the gene expression inference task to demonstrate the advantage of using the TAAFs, the TAAFs are applicable to other tasks outside the gene expression inference, e.g. image compression and reconstruction, forecasting day-ahead electricity prices, and predicting tree height and canopy cover [75]. The gene expression profiling task was chosen for the demonstration as it is a challenging non-linear regression many-to-many task [16]. It was also selected because of its practical relevance to biologists and specialists using the L1000 platform as improving the inference model helps their research. We believe that TAAFs will find use in various fields similar to other adaptive activation functions such as, PReLU [21] and NAF [32]. The TAAFs are especially useful for regression tasks as they mitigate the need to have a linear activation function in the output layer.

## Supporting information

**S1 File. Figure data.** The data necessary for recreating the figures presented in the manuscript.
(XLSX)

## Acknowledgments

Computational resources were supplied by the project "e-Infrastruktura CZ" (e-INFRA LM2018140) provided within the program Projects of Large Research, Development and Innovations Infrastructures. We gratefully acknowledge the support of NVIDIA Corporation for the donation of the Titan Xp GPU used for this research.

## Author Contributions

**Conceptualization:** Vladimír Kunc, Jiří Kléma.

**Formal analysis:** Vladimír Kunc.

**Investigation:** Vladimír Kunc.

**Methodology:** Vladimír Kunc, Jiří Kléma.

**Software:** Vladimír Kunc.

**Supervision:** Jiří Kléma.

**Visualization:** Vladimír Kunc.

**Writing – original draft:** Vladimír Kunc, Jiří Kléma.

**Writing – review & editing:** Vladimír Kunc, Jiří Kléma.

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
