## [Decision Letter · Decision Letter 0]

8 Jul 2020

PONE-D-20-13312

On transformative adaptive activation functions in neural networks for gene expression inference

PLOS ONE

Dear Dr. Kunc,

Thank you for submitting your manuscript to PLOS ONE. After careful consideration, we feel that it has merit but does not fully meet PLOS ONE’s publication criteria as it currently stands. Therefore, we invite you to submit a revised version of the manuscript that addresses the points raised during the review process.

In particular, the authors should address the following points:

1. Please discuss the principal limitations of the D-GEX data, including its probably biased nature. Please explain, how the train-test data split was done. Was any phenotype information considered during that process? If not, please highlight potential biases.

2. Please respond to the individual points raised by both reviewers, specifically regarding the data normalization.

3. The authors highlight TAAF as their main contribution. Please discuss potential applications beyond gene expression inference.

We look forward to receiving your revised manuscript.

Kind regards,

Holger Fröhlich

Academic Editor

PLOS ONE

Journal Requirements:

Reviewers' comments:

Reviewer's Responses to Questions

**Comments to the Author**

1. Is the manuscript technically sound, and do the data support the conclusions?

Reviewer #1: Partly

Reviewer #2: Partly

2. Has the statistical analysis been performed appropriately and rigorously? 

Reviewer #1: Yes

Reviewer #2: N/A

3. Have the authors made all data underlying the findings in their manuscript fully available?

Reviewer #1: Yes

Reviewer #2: Yes

4. Is the manuscript presented in an intelligible fashion and written in standard English?

Reviewer #1: No

Reviewer #2: Yes

5. Review Comments to the Author

Reviewer #1: The authors use transformative adaptive activation functions (TAAFs) in neural networks in order

to predict gene expression values from the L1000 Luminex technology. The main point of the

paper is the improvement of prediction accuracy using their adaptive activation functions. The

authors have made a considerable effort to evaluate their method. I think that the manuscript

could be interesting. However, some major issues should be explained or evaluated:

Major:

1.) The authors mention that they have normalised the gene expression data in a different way, i.e.

by not standardising the expression in a gene wise way. I do agree with their argument that

genuine differences in expression should not be removed by normalisation. However, it makes

the comparison with the D-GEX method difficult. Could the authors test, whether their improvements

using the TAAFs are really a result of the TAAFs, or whether they could have been achieved with

D-GEX and their new normalisation approach.

2.) The scaling parameter beta in Eq. 6 is mathematically redundant. However, the argue that their

experiments show it is helpful. Is this an artefact of the initiation of the weights? What if other methods to initiate the weights are employed? Is this scaling parameter still relevant then? And, if yes, why is this?

3.) Why are the ground truth rankings of the D-GEX method better? I don’t get the argument with the variance. Could the authors check this more carefully and rewrite their explanation?

Minor:

4.) The authors should check their use of articles and prepositions:

Examples:

Line 244: using the NN library keras

Line 262: Replacing tanh by the sigmoid activation function

There are a few more examples. T

5.) There are some typos, e.g

line 221: usage is

Reviewer #2: On transformation adaptive functions in neural networks for gene expression inference

The authors propose an outer transfer function for neural networks, which they use as modification of the original (inner) transfer function. The outer transfer function is called transformative adaptive function (TAAF). They evaluate TAAFs in comparative experiments with an already existing architecture D-GEX on the task of gene expression interference.

####################

My major concern about this work is its general experimental setup. The authors assume a "homogeneous process" of gene expression, which is underlying to all biological cells. This is by far not the case. Gene regulation is (from a biological side) at least dependent on the type of organism, tissue, cell-niche and cell as well as the presence of signalling processes. From a technical perspective, it can additionally be affected by different experimental conditions. The GEO repository comprises data from small comparative experiments (e.g. experimental condition vs control). The corresponding gene expression levels are altered by diseases and/or drugs or even experimentally modified (e.g. knock-down or knock-out experiments).

The aggregation of multiple (GEO-) datasets (as the D-GEX dataset) is therefore likely (or almost certain) to be severely biased and unlikely to reflect the "standard" gene regulation of any cell-type. Samples for the same experimental condition are likely to form self-similar subgroups, which should be easily detectable by any type of cluster algorithm.

This might also affects the "evaluation of the practical impact"as the aggregated samples can not be seen as completely independent samples. As the authors do not control for experimental conditions (or at least do not report it), it is likely that samples from the same experimental conditions are in the training and the test set.

The clustering of the test samples is unnecessary and rather misguiding as the real class structure is given by the experimental conditions of the individual GEO-datasets.

####################

I would expect the chosen normalisation to affect the influence of the individual genes on the proposed scores. Genes with an high expression values are likely to be favoured.

####################

The proposed modification (TAAF) is not motivated and seems to be rather adhoc.

The method introduces for additional parameters (per node). None of them is explained or motivated.

The system is also overparameterized as beta can be formulated as a modification of gamma and the other way round.

6. PLOS authors have the option to publish the peer review history of their article (what does this mean?). If published, this will include your full peer review and any attached files.

Reviewer #1: No

Reviewer #2: No

---

## [Author Response · Author response to Decision Letter 0]

31 Oct 2020

Dear Prof. Fröhlich,

thank you for inviting us to submit a revised draft of our manuscript entitled “On transformative adaptive activation functions in neural networks for gene expression inference”. Now we resubmit our article for further consideration. In the revised manuscript, we have carefully considered reviewers’ comments and suggestions. As instructed, we have attempted to explain the changes made in reaction to all the comments. We reply to each comment in point-by-point way. 

The reviewers’ comments were very helpful and constructive. To fully answer the reviewers’ concerns we performed extensive experiments and took the extra time to finish them. After addressing the issues raised, we believe that the quality of our paper has significantly improved.

Sincerely,

Vladimír Kunc, Jiří Kléma

Editor suggestions

1. Please discuss the principal limitations of the D-GEX data, including its probably biased nature. Please explain, how the train-test data split was done. Was any phenotype information considered during that process? If not, please highlight potential biases.

The D-GEX data contain more that 100,000 samples. This extent makes the data unique and truly representative. It also helps to sufficiently fit the neural network models. On the other hand, such a representative gene expression dataset necessarily contains a heterogeneous sample set (different GEO series, laboratories and phenotypes). In the original manuscript, we performed random train-test sample splits, i.e., the splits that did not consider the available sample annotation. We did it in the same way as the authors of the original D-GEX and the following studies did. Because our experiments are comparative and large-scale, we worked with the assumption that the potential biases cannot directly influence them.

We agree with the reviewers that this justification is rather vague; we did it implicitly only too. In the revised manuscript, we carried out completely new experiments with heterogeneity-aware splits. These splits take the sample annotation into consideration. In particular, no GEO-series could appear both in the train and test data at the same time.

These splits minimize the chance that the same experimental conditions appear in the training and the test set. At the same time, it is a split procedure that is feasible for such a large amount of data. The results reached proved to be unaffected by the potential learning biases

resulting from this type of heterogeneity in the input dataset.

2. Please respond to the individual points raised by both reviewers, specifically regarding the data normalization.

We respond to all the points raised by both reviewers below. The data normalization issue is addressed as well. In particular, we reformulated and extended the text that discusses this issue.

3. The authors highlight TAAF as their main contribution. Please discuss potential applications beyond gene expression inference.

The proposal of transformative adaptive activation functions makes our main contribution. In the original manuscript, the contribution was experimentally justified. In the revised manuscript, we extend the experimental justification (the bias issues, the further verification of its practical impact). In the modified section “Experiment 4: Importance of individual parameters”, we also generally discuss the positive role of redundancy in neural networks and put TAAFs in a wider context. The main idea is that the overspecification in TAAFs causes the gradient descent procedures used to learn the networks less likely to get stuck in poor local minima and plateaus. For this reason, we assume that TAAFs are generally applicable and can be tested in domains such as image compression and reconstruction, natural language processing, or fraud detection. However, the actual applicability of the method wrt the dataset characterization (the dataset size and its further information/statistical properties) needs to be further analyzed and we would only speculate about it at the moment. We believe that the experimentally bold result that we reached for a representative and large gene expression dataset is significant both for the biological and AI community.

Reviewer 1

1. The authors mention that they have normalised the gene expression data in a different way, i.e. by not standardising the expression in a gene wise way. I do agree with their argument that genuine differences in expression should not be removed by normalisation. However, it makes the comparison with the D-GEX method difficult. Could the authors test, whether their improvements using the TAAFs are really a result of the TAAFs, or whether they could have been achieved with D-GEX and their new normalisation approach.

We have re-implemented the original D–GEX and trained it on the same data as the modified D–GEX with the novel TAAFs was trained on. Both models’ performances are reported on the same data with the same normalization; thus any differences in the reported performance are due only to the model differences. We have clarified that in the manuscript in the section Material and methods.

2. The scaling parameter beta in Eq. 6 is mathematically redundant. However, the argue that their experiments show it is helpful. Is this an artefact of the initiation of the weights? What if other methods to initiate the weights are employed? Is this scaling parameter still relevant then? And, if yes, why is this?

The scaling parameter β is initialized in such a way that it has no influence initially — it is initialized to 1. Only during the training, the parameter is changed, thus since the experiments have shown that it improves the network performance, it has to be useful for the training procedure. We have extended the section discussing the possible reasons why it might be improving the performance despite the apparent redundancy. One of the hypotheses, why it might be beneficial, is that its inclusion modifies the optimization landscape in a way making it easier to navigate to a good optimum: "higher dimensions also means more potential directions of descent, so perhaps the gradient descent procedures used in practice are more unlikely to get stuck in poor local minima and plateaus" [1].

While other initialization procedures for the added parameters might be (and probably will be) beneficial, we have set up the initialization such that the shape of the inner activation function is the same as if there were no additional parameters (i.e., the scaling parameters are set to 1, the translation parameters to 0) to keep the starting point the same as for the

original D-GEX without the adaptive functions. Different initialization methods are part of planned future research.

3. Why are the ground truth rankings of the D-GEX method better? I don’t get the argument with the variance. Could the authors check this more carefully and rewrite their explanation?

We agree that the original text was misleading in this aspect; we have clarified the text, in particular, we extended the section Practical impact on differential gene expression analysis.

What we meant is that the D–GEX with TAAFs has better accuracy when identifying the differentially expressed genes (compared to the plain D–GEX) because it captures more details in the data (i.e., explains more variance present in the data). Therefore it can systematically produce better rankings of the differentially expressed genes candidates even for large sample sizes because it represents the inferred expressions more faithfully and thus the MAEs for both methods do not have to converge to a common number with increasing sample size. We have rephrased the section to make it clearer.

4. The authors should check their use of articles and prepositions:

Examples:

Line 244: using the NN library keras

Line 262: Replacing tanh by the sigmoid activation function

Thank you for the corrections. We went over the text carefully and addressed such issues.

The revised manuscript was proofread by a native speaker.

5. There are some typos, e.g

line 221: usage is

We went over the text carefully and addressed such issues.

Reviewer 2

1. My major concern about this work is its general experimental setup. The authors assume a "homogeneous process" of gene expression, which is underlying to all biological cells. This is by far not the case. Gene regulation is (from a biological side) at least dependent on the type of organism, tissue, cell-niche and cell as well as the presence of signalling processes. From a technical perspective, it can additionally be affected by different experimental conditions. The GEO repository comprises data from small comparative experiments (e.g. experimental condition vs control). The corresponding gene expression levels are altered by diseases and/or drugs or even experimentally modified (e.g. knock-down or knock-out experiments).

The aggregation of multiple (GEO-) datasets (as the D-GEX dataset) is therefore likely (or almost certain) to be severely biased and unlikely to reflect the "standard" gene regulation of any cell-type. Samples for the same experimental condition are likely to form self-similar subgroups, which should be easily detectable by any type of cluster algorithm.

We agree that the gene regulation is dependent on many factors while some of the factors are not available in the present data; however, it is the assumption behind the L1000 platform that these factors should reflect on the expression of the landmark genes and thus indirectly reflect in the gene expressions reconstructed using the landmark genes. It is certain that some of the factors are not reflected entirely (or at all) in the expression of the landmark genes and thus the expression inference will not be perfect; nevertheless, the LINCS program still found it worthwhile to develop the L1000 platform capturing only the selected landmark genes and to infer the expression of other genes.

We also agree that the aggregation of multiple GEO- datasets does not reflect the "standard" gene regulation; however, it is the only data that are available for fitting such models and, to our best knowledge, all studies inferring the gene expressing from the L100 platform use the D–GEX dataset in particular [2]–[5]. On the other hand, the inclusion of such experiments where the gene expression levels are altered by diseases and other factors in the dataset makes the L1000 platform to be usable on these kinds of experiments as well as such experiments were taken into account when training the models (which would not be the case if the model was trained only on samples reflecting the "standard" gene regulation).

While we agree that most of the models present in the literature for inferring the gene expression levels from the L1000 platform are biased due to the usage of the aggregated GEO- data for model fitting, our main contribution that given the data, the D–GEX with TAAFs performed significantly better than the plain D–GEX.

2. This might also affects the "evaluation of the practical impact"as the aggregated samples can not be seen as completely independent samples. As the authors do not control for experimental conditions (or at least do not report it), it is likely that samples from the same experimental conditions are in the training and the test set.

Originally, we performed random train-test sample splits, i.e, the splits that did not consider the available sample annotation. This is the way the authors of the original D-GEX and the subsequent studies took. Since our experiments are comparative and large-scale, we worked with the assumption that the potential biases cannot directly influence them.

We agree with you that such a justification is rather vague, we did it implicitly only too. In the revised manuscript, we added another version of the dataset used with heterogeneity-aware train-test splits and carried additional experiments using such data to show that the reported results are not affected by the potential biases resulting from this type of heterogeneity in the input dataset.

The added dataset is based on the original D–GEX dataset, this time we split it in such a way that no samples from a single GEO- series are both in the train set and test set, i.e., if a sample is from one particular series, all other samples from the same series have to be in the same split to prevent leakages between the training and the test set. While there still might be samples from similar tissues and/or experiments following similar protocols in both the training and the sets, this should prevent that these will be samples from the same experiment. Using this dataset, we ran additional differential gene expression tasks to show that the performance difference between the models is due to the TAAFs and not due to the potential bias caused inclusion of samples from the same experiment in both training and testing sets.

While grouping the samples by their GEO- series will not guarantee perfect independence of samples as there might be, for example, slightly different procedures and equipment used by groups of laboratories leading to some clusters between samples, it should be sufficient as the most dangerous dependence is mitigated and more detailed approaches would require manual curation of the samples and their supplementary information to identify possible dependencies and clusters which would be a very demanding task for a dataset of this size.

3. The clustering of the test samples is unnecessary and rather misguiding as the real class structure is given by the experimental conditions of the individual GEO-datasets.

We agree that usage of the phenotype data from the individual GEO-datasets is a better option and we amended the manuscript by adding a set of experiments using such data. We kept the clustering-based experiment as it allowed for larger sample sizes since the largest GEO- series available in the original D-GEX dataset had classes with fewer examples. Still, we amended it by a set of experiments using the phenotype data from the series.

We chose the GSE2109 series and used the individual tissues as the classes. We used all classes that had more than 100 samples (which were 6 tissue types) and ran the evaluation of practical impact using the DGE analysis for all tissue pairs (15 pairs). We thank you for the suggestion as we believe that the real phenotype data will show the difference better than using just the artificial task based on the clustering. We showed that while there are differences in the numbers of differentially expressed genes and the differences in the estimation difficulty, the D-GEX with the proposed TAAFs performed better than the plain D-GEX for all task pairs and most sample sizes for all measured metrics (accuracy, Matthews correlation coefficient, F0.5, F1, and F2 scores) and also produced better candidate rankings (in terms of MAE based on the ranks) as shown in Figs. 11–14 in the manuscript.

4. I would expect the chosen normalisation to affect the influence of the individual genes on the proposed scores. Genes with an high expression values are likely to be favoured.

There is no perfect way to normalize the data — either the genes with expression near the noise levels will be relative favored (i.e., will have the same importance as other genes despite being mostly noise) or the genes with high expressions will be favored. The author of the original D-GEX used the former, we have opted for the latter as we have observed that the former method of normalization makes the models focusing too much on the genes with near noise expression levels at the expense of inference quality of expression of other genes and also loses the information of the relative differences in the expression levels between the genes. While both approaches have their cons and pros, we think that the latter approach is more of a use for biologists and other specialists as they are usually interested in genes with expressions higher than the noise level.

There are some approaches that are a combination of the two previous approaches that could, to some extent, mitigate the mentioned issues by using non-linear weight based on the expression levels but we have felt that analyzing different weighting approaches would be greatly out-of-the-scope of the work. Furthermore, the data were already non-linearly transformed due to the preprocessing of the raw microarray data and then the quantile normalization we have performed, thus to investigate various weighting approaches we would have to also thoroughly investigate the influence of different preprocessing of the raw microarray data on the trained model performance which would be worth of another paper. Normalization proved to have a negligible impact on our comparative experiments.

5. The proposed modification (TAAF) is not motivated and seems to be rather adhoc.

The method introduces for additional parameters (per node). None of them is explained or motivated.

Thank you for your comment, we clarified this in the revised manuscript in the section Proposed transformative adaptive activation function. Briefly summarized, the motivation behind the individual parameters is that they allow for arbitrary translation and scaling of the activation functions. For example, the translation and scaling is the only difference between two popular activation functions — hyperbolic tangent and logistic sigmoid — and sometimes one outperforms the other depending on the model and the data. Our proposed activation function mitigates the need for the choice as it is able to freely transform between both of the functions (and more). It was also shown in the literature that activation functions with only one of the introduced parameters are significantly helping the training (e.g., the trainable amplitude) — our work is an extension of already established adaptive activation functions that were too constrained or did not allow for some of the adaptions the proposed TAAFs do. The motivation is therefore to allow the adaptation of the activation function to the given task by allowing for arbitrary translation and scaling.

Furthermore, such an adaptive activation function removes the need to have a linear activation function in the last layer for regression tasks as is usually done. The usage of the linear function in the last layer requires to have a full set of weights for the incoming connections just for the ability to scale the output to an arbitrary range while the proposed TAAF can do it with only 4 additional parameters (both approaches are different in nature, the linear activation function leads to a weighted sum of the incoming signals while the TAAF applies a non-linear transformation as well). We have shown that replacing the linear activation functions in the last layer was beneficial to the performance of the D-GEX.

6. The system is also overparameterized as beta can be formulated as a modification of gamma and the other way round.

The reviewer is right that the system is overparameterized — β can be factored into the weights wi (not into the γ — β is a multiplicative parameter while γ is an additive parameter).

While we have originally only shown that the inclusion of the parameter β is beneficial despite the apparent redundancy, we have extended the relevant section by discussing how the apparently redundant parameter β might help to the optimization of the model. We also newly refer to resources discussing similar forms of redundancy and their benefits for artificial neural network optimization. One of the hypotheses, why the parameter β might be beneficial, is that its inclusion modifies the optimization landscape in a way making it easier to navigate to a good optimum: "higher dimensions also means more potential directions of

descent, so perhaps the gradient descent procedures used in practice are more unlikely to get stuck in poor local minima and plateaus" [1].

References

[1] I. Safran and O. Shamir, “On the quality of the initial basin in overspecified neural networks”, M. F. Balcan and K. Q. Weinberger, Eds., ser. Proceedings of Machine Learning Research, vol. 48, New York, New York, USA: PMLR, 20–22 Jun 2016, pp. 774–782. [Online]. Available: http://proceedings.mlr.press/v48/safran16.html.

[2] Y. Chen et al., “Gene expression inference with deep learning”, Bioinformatics, vol. 32, no. 12, pp. 1832–1839, Feb. 2016. doi: 10 . 1093 / bioinformatics / btw074. [Online]. Available: https://doi.org/10.1093/bioinformatics/btw074.

[3] X. Wang, K. G. Dizaji, and H. Huang, “Conditional generative adversarial network for gene expression inference”, Bioinformatics, vol. 34, no. 17, pp. i603–i611, Sep. 2018. doi: 10.1093/bioinformatics/bty563. [Online]. Available: https://doi.org/10.1093/bioinformatics/bty563.

[4] H. Wang, C. Li, J. Zhang, J. Wang, Y. Ma, and Y. Lian, “A new LSTM-based gene expression prediction model: L-GEPM”, Journal of Bioinformatics and Computational Biology, vol. 17, no. 04, p. 1 950 022, Aug. 2019. doi: 10 . 1142 / s0219720019500227. [Online]. Available: https://doi.org/10.1142/s0219720019500227.

[5] K. G. Dizaji, X. Wang, and H. Huang, “Semi-supervised generative adversarial network for gene expression inference”, in Proceedings of the 24th ACM SIGKDD International Conference on Knowledge Discovery & Data Mining - KDD’18, New York: ACM Press, 2018. doi: 10.1145/3219819.3220114. [Online]. Available: https://doi.org/10.1145/3219819.3220114.

---

## [Decision Letter · Decision Letter 1]

1 Dec 2020

On transformative adaptive activation functions in neural networks for gene expression inference

PONE-D-20-13312R1

Dear Dr. Kunc,

We’re pleased to inform you that your manuscript has been judged scientifically suitable for publication and will be formally accepted for publication once it meets all outstanding technical requirements.

Kind regards,

Holger Fröhlich

Academic Editor

PLOS ONE

Additional Editor Comments (optional):

Reviewers' comments:

Reviewer's Responses to Questions

**Comments to the Author**

1. If the authors have adequately addressed your comments raised in a previous round of review and you feel that this manuscript is now acceptable for publication, you may indicate that here to bypass the “Comments to the Author” section, enter your conflict of interest statement in the “Confidential to Editor” section, and submit your "Accept" recommendation.

Reviewer #1: All comments have been addressed

2. Is the manuscript technically sound, and do the data support the conclusions?

Reviewer #1: Yes

3. Has the statistical analysis been performed appropriately and rigorously? 

Reviewer #1: Yes

4. Have the authors made all data underlying the findings in their manuscript fully available?

Reviewer #1: Yes

5. Is the manuscript presented in an intelligible fashion and written in standard English?

Reviewer #1: Yes

6. Review Comments to the Author

Reviewer #1: All my previous concerns have been convincingly addressed. I think the manuscript could be published after checking for some typos and some minor language editing.

7. PLOS authors have the option to publish the peer review history of their article (what does this mean?). If published, this will include your full peer review and any attached files.

Reviewer #1: No

---

## [Editor Report · Acceptance letter]

4 Jan 2021

PONE-D-20-13312R1 

On transformative adaptive activation functions in neural networks for gene expression inference 

Dear Dr. Kunc:

I'm pleased to inform you that your manuscript has been deemed suitable for publication in PLOS ONE. Congratulations! Your manuscript is now with our production department. 

Kind regards, 

on behalf of

Prof. Dr. Holger Fröhlich 

Academic Editor

PLOS ONE